



**Inflammatory responses to secondary organic aerosols (SOA) generated from biogenic and**
**anthropogenic precursors**
*Wing Y. Tuet[1], Yunle Chen[2], Shierly Fok[1], Julie A. Champion[1], Nga L. Ng[1,3*]*
[1]School of Chemical and Biomolecular Engineering, Georgia Institute of Technology, Atlanta, GA
[2]School of Materials Science and Engineering, Georgia Institute of Technology, Atlanta, GA
[3]School of Earth and Atmospheric Sciences, Georgia Institute of Technology, Atlanta, GA
**Corresponding Author**
[*]email: ng@chbe.gatech.edu
Keywords: reactive oxygen/nitrogen species, inflammatory cytokines, particulate matter, secondary
organic aerosol





Abstract

12        Cardiopulmonary health implications resulting from exposure to secondary organic

aerosols (SOA), which comprise a significant fraction of ambient particulate matter (PM), have
received increasing interest in recent years. In this study, alveolar macrophages were exposed to
SOA generated from the photooxidation of biogenic and anthropogenic precursors (isoprene, α-
pinene, β-caryophyllene, pentadecane, *m*-xylene, and naphthalene) under different formation
conditions ($RO_2$ + $HO_2$ vs. $RO_2$ + NO dominant, dry vs. humid). Various cellular responses were
measured, including reactive oxygen/nitrogen species (ROS/RNS) production and secreted levels
of cytokines, tumor necrosis factor-α (TNF-α) and interleukin-6 (IL-6). SOA precursor identity
and formation condition affected all measured responses in a hydrocarbon-specific manner. With
the exception of naphthalene SOA, cellular responses followed a trend where TNF-α levels
reached a plateau with increasing IL-6 levels. ROS/RNS levels were consistent with relative levels
of TNF-α and IL-6, due to their respective inflammatory and anti-inflammatory effects. Exposure
to naphthalene SOA, whose aromatic ring-containing products may trigger different cellular
pathways, induced higher levels of TNF-α and ROS/RNS than suggested by the trend. Distinct
cellular response patterns were identified for hydrocarbons whose photooxidation products shared
similar chemical functionalities and structures, which suggests that the carbon backbone may be
important for determining cellular effects. A positive nonlinear correlation was also detected
between ROS/RNS levels and previously measured DTT activities for SOA samples. In the context
of ambient samples collected during summer and winter in the greater Atlanta area, all laboratory-
generated SOA produced similar or higher levels of ROS/RNS and DTT activities. These results
suggest that the health effects of SOA are important considerations for understanding the health
implications of ambient aerosols.



## Introduction

Particulate matter (PM) exposure is a leading global risk factor for human health (Lim et al., 2012) with numerous studies reporting associations between elevated PM concentrations and increases in cardiopulmonary morbidity and mortality (Li et al., 2008; Pope III and Dockery, 2006; Brunekreef and Holgate, 2002; Dockery et al., 1993; Hoek et al., 2013; Anderson et al., 2011; Pope et al., 2002). A possible mechanism for PM-induced health effects has been suggested by toxicology studies, wherein PM-induced oxidant production, including reactive oxygen and nitrogen species (ROS/RNS), initiates inflammatory cascades thus resulting in oxidative stress and cellular damage (Li et al., 2003a; Tao et al., 2003; Castro and Freeman, 2001; Gurgueira et al., 2002; Wiseman and Halliwell, 1996; Hensley et al., 2000). Furthermore, prolonged stimulation of these inflammatory cascades may lead to chronic inflammation, for which there is a recognized link to cancer (Philip et al., 2004). Together, these findings suggest that a possible relationship exists between PM exposure and observed health effects.

Various assays have been developed to study PM-induced oxidant production, including cell-free chemical assays that measure the oxidative potential of PM samples (Kumagai et al., 2002; Cho et al., 2005; Fang et al., 2015b) as well as cellular assays that measure intracellular ROS/RNS produced as a result of PM exposure (Landreman et al., 2008; Tuet et al., 2016). Cell-free assays simulate biologically relevant redox reactions using an anti-oxidant species (e.g. dithiothreitol, DTT; ascorbic acid, AA). The anti-oxidant is oxidized via electron transfer reactions catalyzed by redox-active species in the PM sample and its rate of decay serves as a measure of the concentration of redox-active species present (Fang et al., 2015b). On the other hand, cellular assays utilize a fluorescent probe (e.g. carboxy-$H_2DCFDA$) that reacts with ROS/RNS and the measured fluorescence is proportional to the concentration of ROS/RNS produced as a result of





PM exposure (Landreman et al., 2008; Tuet et al., 2016). Both types of assays have been utilized
extensively to characterize a variety of PM samples and identify sources that may be detrimental
to health (Verma et al., 2015a; Saffari et al., 2015; Fang et al., 2015a; Bates et al., 2015; Li et al.,
2003b; Tuet et al., 2016). In particular, numerous studies suggest that organic carbon constituents,
especially humic-like substances (HULIS) and oxygenated polyaromatic hydrocarbons (PAH),
may contribute significantly to PM-induced oxidant production (Li et al., 2003b; Kleinman et al.,
2005; Hamad et al., 2015; Verma et al., 2015b; Lin and Yu, 2011). Furthermore, recent
measurements of ROS/RNS production and DTT activity using ambient samples collected in
summer and winter around the greater Atlanta area showed that there is a significant correlation
between summertime organic species and intracellular ROS/RNS production, suggesting a
possible role for secondary organic aerosols (SOA) (Tuet et al., 2016). The same study also
reported a significant correlation between ROS/RNS production and DTT activity for summer
samples, while a relatively flat ROS/RNS response was observed for winter samples spanning a
similar DTT activity range (Tuet et al., 2016). These results highlight a need to consider multiple
endpoints as a simple correlation may not exist between different endpoints, especially cellular
responses that may result from complicated response networks.
Despite these findings, there are still many gaps in knowledge regarding PM-induced
health effects. While field studies repeatedly showed that SOA often dominate over primary
aerosols even in urban environments (Zhang et al., 2007; Jimenez et al., 2009; Ng et al., 2010),
many prior health studies have focused on the effects of primary emissions (e.g. PM emitted
directly from combustion engines) (Kumagai et al., 2002; Bai et al., 2001; McWhinney et al.,
2013a; Turner et al., 2015) rather than those of SOA formed from the oxidation of emitted
hydrocarbons (McWhinney et al., 2013b; Rattanavaraha et al., 2011; Kramer et al., 2016; Lund et





al., 2013; McDonald et al., 2010; McDonald et al., 2012; Baltensperger et al., 2008; Arashiro et
al., 2016; Platt et al., 2014). The cellular exposure studies that do explore SOA focused on SOA
formed from a single SOA precursor and include different measures of response (e.g. ROS/RNS,
inflammatory biomarkers, gene expression, etc.) (Arashiro et al., 2016; Lund et al., 2013;
McDonald et al., 2010; McDonald et al., 2012; Baltensperger et al., 2008). As a result, there is a
lack of understanding in terms of the relative toxicity of individual SOA systems. Recently, Tuet
et al. (2017) systematically investigated the DTT activities of SOA formed from different biogenic
and anthropogenic precursors and demonstrated that intrinsic DTT activities were highly
dependent on SOA precursor identity, with naphthalene SOA having the highest DTT activity. As
a result, a systematic study on the cellular responses induced by these SOA systems may provide
similar insights. Furthermore, cellular responses may complement these previously measured DTT
activities to elucidate a more complete picture of the health effects of PM.
In the present study, alveolar macrophages were exposed to SOA generated under different
formation conditions from various SOA precursors. Cellular responses induced by SOA exposure
were measured, including intracellular ROS/RNS production and levels of tumor necrosis factor-
$\alpha$ (TNF-$\alpha$) and interleukin-6 (IL-6). Intracellular ROS/RNS production serves as a general
indicator of oxidative stress, whereas TNF-$\alpha$ and IL-6 are pro-inflammatory cytokines indicative
of the inflammatory response (Henkler et al., 2010; Kishimoto, 2003; Wang et al., 2003).
Precursors were chosen to include major classes of biogenic and anthropogenic compounds known
to produce SOA upon atmospheric oxidation (Table S1). The selected biogenic precursors include:
isoprene, the most abundant non-methane hydrocarbon (Guenther et al., 2006); $\alpha$-pinene, a well-
studied monoterpene with emissions on the order of global anthropogenic emissions (Guenther et
al., 1993; Piccot et al., 1992); and $\beta$-caryophyllene, a representative sesquiterpene. Both



monoterpenes and sesquiterpenes have been shown to contribute significantly to ambient aerosol
(Eddingsaas et al., 2012; Hoffmann et al., 1997; Tasoglou and Pandis, 2015; Goldstein and
Galbally, 2007). Similarly, the anthropogenic precursors include: pentadecane, a long-chain
alkane; *m*-xylene, a single-ring aromatic; and naphthalene, a poly-aromatic. These compounds are
emitted as products of incomplete combustion (Robinson et al., 2007; Jia and Batterman, 2010;
Bruns et al., 2016) and have considerable SOA yields (Chan et al., 2009; Ng et al., 2007b; Lambe
et al., 2011). In addition to precursor identity, the effects of humidity (dry vs. humid) and $NO_x$
levels (different predominant peroxy radical ($RO_2$) fates, $RO_2 + HO_2$ vs. $RO_2 + NO$) on SOA
cellular inflammatory responses were investigated, as different formation conditions have been
shown to affect aerosol chemical composition and mass loading, which could in turn result in a
different cellular response (Chhabra et al., 2010; Chhabra et al., 2011; Eddingsaas et al., 2012; Ng
et al., 2007b; Loza et al., 2014; Ng et al., 2007a; Chan et al., 2009; Boyd et al., 2015). Finally,
correlations between bulk aerosol composition, specifically elemental ratios, and cellular
inflammatory responses were investigated to determine whether there is a link between different
inflammatory responses and aerosol composition.
Methods
**Alveolar macrophage cell line.** Immortalized murine alveolar macrophages (MH-S,
ATCC®CRL-2019™) were cultured in RPMI-1640 media supplemented with 10% fetal bovine
serum (FBS, Quality Biological, InC.), 1% penicillin-streptomycin, and 50 µM β-mercaptoethanol
(BME) at 37°C and humid air containing 5% $CO_2$. For exposure experiments, MH-S cells were
seeded at a density of 2 x $10^4$ cells well$^{-1}$ onto 96-well plates pre-treated with 10% FBS in
phosphate buffered saline (PBS, Cellgro). For seeding and all assay procedures thereon, FBS-





supplemented cell culture media without BME addition was used as BME is a reducing agent that
may interfere with inflammatory measurements.
**Chamber experiments.** SOA formed form the photooxidation of biogenic and
anthropogenic precursors were generated in the Georgia Tech Environmental Chamber (GTEC)
facility. Details of the facility have been described elsewhere (Boyd et al., 2015). Briefly, the
chamber facility consists of two 12 $m^3$ Teflon chambers suspended within a 21 x 12 ft temperature-
controlled enclosure. Black lights and natural sunlight fluorescent lamps surround the chambers,
and multiple sampling ports allow for injection of reagents, as well as gas- and aerosol-phase
measurements. Gas-phase $O_3$, $NO_2$, and $NO_x$ concentrations were monitored using an $O_3$ analyzer
(Teledyne T400), a cavity attenuated phase shift (CAPS) $NO_2$ monitor (Aerodyne), and a
chemiluminescence $NO_x$ monitor (Teledyne 200EU) respectively, while hydrocarbon decay was
monitored using a gas chromatography-flame ionization detector (GC-FID, Agilent 7890A).
Hydrocarbon decay was also used to estimate hydroxyl radical (OH) concentrations. For aerosol-
phase measurements, a Scanning Mobility Particle Sizer (SMPS, TSI) was used to measure aerosol
volume concentrations and distributions, while a High Resolution Time-of-Flight Aerosol Mass
Spectrometer (HR-ToF-AMS, Aerodyne; henceforth referred to as the AMS) was used to
determine bulk aerosol composition (DeCarlo et al., 2006). AMS data was analyzed using the data
analysis toolkit SQUIRREL (v. 1.57) and PIKA (v. 1.16G). Elemental ratios, including O:C, H:C,
and N:C, were obtained using the method outlined by Canagaratna et al. (2015) and used to
calculate the average carbon oxidation state ($\overline{OS}_c$) (Kroll et al., 2011). Temperature and relative
humidity (RH) were also monitored using a hydro-thermometer (Vaisala HMP110).
Experiments were designed to probe the effects of humidity, $RO_2$ fate, and precursor
identity on cellular inflammatory responses induced by different SOA formed under these





conditions (Table 1). All chamber experiments were performed at ~25 °C under dry (RH < 5%) or
humid (RH ~ 45%) conditions. Chambers were flushed with pure air for ~24 hrs prior to each
experiment. During this time, chambers were also humidified for humid experiments by means of
a bubbler filled with deionized (DI) water. Seed aerosol was injected by atomizing a 15 mM
$(NH_4)_2SO_4$ seed solution (Sigma Aldrich) to obtain a seed concentration of ~20 µg m$^{-3}$. It should
be noted that experimental conditions deviate for experiment 7 (isoprene SOA under $RO_2 + HO_2$
dominant, "humid" conditions) due to low SOA mass yields. For this experiment, an acidic seed
solution (8 mM $MgSO_4$ and 16 mM $H_2SO_4$) and a dry chamber were used to promote SOA
formation via the isoprene epoxydiol (IEPOX) uptake pathway. This pathway has been shown to
contribute significantly to ambient OA and has a higher SOA mass yield compared to the IEPOX
+ OH pathway (Surratt et al., 2010; Lin et al., 2012; Xu et al., 2015).

SOA precursor was then introduced by injecting a known amount of hydrocarbon solution

[isoprene, 99%; α-pinene, ≥ 99%; β-caryophyllene, > 98.5%; pentadecane, ≥ 99%; *m*-xylene, ≥
99%; naphthalene, 99% (Sigma Aldrich)] into a glass injection bulb and passing zero air over the
solution until it fully evaporated. For pentadecane and β-caryophyllene, the glass bulb was also
heated gently during hydrocarbon injection to ensure full evaporation (Tasoglou and Pandis,
2015). Naphthalene was injected by passing zero air over solid naphthalene flakes as described in
previous studies (Chan et al., 2009). OH precursor was then introduced via injection of hydrogen
peroxide ($H_2O_2$) for $RO_2 + HO_2$ experiments or nitrous acid (HONO) for $RO_2 + NO$ experiments.
For $H_2O_2$, a 50% aqueous solution (Sigma Aldrich) was injected using the same method described
for hydrocarbon injection to achieve an $H_2O_2$ concentration of 3 ppm. This amount yielded OH
concentrations on the order of $10^6$ molec cm$^{-3}$. For HONO injections, HONO was first prepared
by adding 10 mL of 1%wt aqueous $NaNO_2$ (VWR International) dropwise into 20 mL of 10%wt





$H_2SO_4$ (VWR International) in a glass bulb. Zero air was then passed over the solution to introduce
HONO into the chamber (Chan et al., 2009; Kroll et al., 2005). Photolysis of HONO yielded OH
concentrations on the order of $10^7$ molec cm$^{-3}$. NO and $NO_2$ were also formed as byproducts of
HONO synthesis. Once all the $H_2O_2$ evaporated ($RO_2 + HO_2$ experiments) or $NO_x$ concentrations
stabilized ($RO_2 + NO$ experiments), the UV lights were turned on to initiate photooxidation.
**Aerosol collection and extraction.** Aerosol samples were collected onto 47 mm Teflon$^{TM}$
filters (0.45 µm pore size, Pall Laboratory). The total mass collected onto each filter was
determined by integrating the SMPS time-dependent volume concentration over the filter
collection period and multiplying by the total volume of air collected. To account for potential
$H_2O_2$ or HONO uptake, background filters were also collected. These filters were collected when
only seed particles and OH precursor ($H_2O_2$ or HONO) were injected into the chamber under
otherwise identical experimental conditions. All collected samples were placed in sterile petri
dishes, sealed with Parafilm M®, and stored at -20 °C until extraction and analysis (Fang et al.,
2015b). Collected particles were extracted following the procedure outlined in Fang et al. (2015a)
with modifications for cellular exposure. Briefly, filter samples were submerged in cell culture
media (RPMI-1640) and sonicated for two 30 min intervals (1 hr total) using an Ultrasonic
Cleanser (VWR International). In between sonication intervals, the water was replaced to reduce
bath temperature. After the final sonication interval, sample extracts were filtered using 0.45 µm
PTFE syringe filters (Fisherbrand$^{TM}$) to remove any insoluble material and supplemented with 10%
FBS (Fang et al., 2015b).
**Intracellular ROS/RNS measurement.** ROS/RNS were detected using the assay
optimized in Tuet et al. (2016). Briefly, the assay consists of five major steps: (1) pre-treatment of
96-well plates to ensure a uniform cell density, (2) seeding of cells onto pre-treated wells at 2 x



$10^4$ cells well$^{-1}$, (3) incubation with ROS/RNS probe (carboxy-H$_2$DCFDA, Molecular Probes C-
400) diluted to a final concentration of 10 µM, (4) exposure of probe-treated cells to samples and
controls for 24 hrs, and (5) detection of ROS/RNS using a microplate reader (BioTek Synergy H4,
ex/em: 485/525 nm). Positive controls included bacterial cell wall component lipopolysaccharide
(LPS, 1 µg mL$^{-1}$), H$_2$O$_2$ (100 µM), and reference filter extract (10 filter punches mL$^{-1}$, 1 per filter
sample, from various ambient filters collected at the Georgia Tech site, while negative controls
included blank filter extract (2 punches mL$^{-1}$) and control cells (probe-treated cells exposed to
media only, no stimulants).
A previous study on the ROS/RNS produced induced by exposure to ambient PM samples
found that ROS/RNS production was highly dose-dependent and could therefore not be
represented by measurements taken at a single dose (Tuet et al., 2016). Here, we utilize the dose-
response curve approach described in Tuet et al. (2016). For each aerosol sample, ROS/RNS
production was measured over ten dilutions and expressed as a fold increase in fluorescence over
control cells. A representative dose-response curve is shown in Fig. 1. For comparisons to other
inflammatory endpoints and chemical composition, ROS/RNS production was represented using
the area under the dose-response curve (AUC), as AUC has been shown to be the most robust
metric for comparing PM samples (Tuet et al., 2016).
**Cytokine measurement.** Secreted levels of TNF-α and IL-6 were measured post-exposure
(24 hrs) using enzyme-linked immunosorbent assay (ELISA) kits following manufacturer's
specifications (ThermoFisher). All measurements were carried out using undiluted cell culture
supernatant. For each aerosol sample, TNF-α and IL-6 were measured over seven dilutions and
represented as a fold increase over control. Similarly, the AUC was used to represent each endpoint
for comparison purposes.



**Cellular metabolic activity.** The MTT (3-(4,5-dimethylthiazol-2-yl)-2,5-
diphenyltetrazolium bromide) assay (Biotium) was used to assess cellular metabolic activity.
Briefly, supernatants containing sample extracts were removed after the exposure period and
replaced with media containing MTT. Cells were then returned to the incubator for 4 hrs, during
which the tetrazolium dye was reduced by cellular NAD(P)H-dependent oxidoreductases to
produce an insoluble purple salt (formazan). Dimethyl sulfoxide was then used to solubilize the
salt and the absorbance at 570 nm was determined using a microplate reader (BioTek Synergy H4).
**Statistical analysis.** Linear regressions between bulk aerosol composition and cellular
inflammatory responses were evaluated using Pearson's correlation coefficient, and the
significance of each correlation coefficient was determined using multiple imputation, which
calculated the total variance associated with the slope of each regression. Details of this method
are described in Pan and Shimizu (2009). Briefly, response parameters (i.e. AUCs for each
endpoint) were assumed to follow a normal distribution. Ten "estimates" were obtained for each
response using the average and standard deviation determined from the dose-response curve fit.
These estimates were then plotted against bulk aerosol composition (e.g. O:C, H:C, and N:C) to
obtain ten fits, and the slopes and variances generated from these fits were used to calculate the
between and within variance. Finally, a Student's *t*-test was used to calculate and evaluate the
associated *p*-values using a 95% confidence interval.
Results and Discussion
**Effect of SOA precursor and formation condition on SOA inflammatory response.** To
investigate whether SOA formed from different precursors elicited different inflammatory
responses, levels of ROS/RNS, TNF-α, and IL-6 were measured after exposing alveolar





macrophages to SOA generated from six VOCs generated under three formation conditions (Table
1). The AUC per mass of SOA (μg) in the extract for ROS/RNS, TNF-α and IL-6 are shown in
Fig. 2, shaped by SOA formation condition. It should be noted that all responses were normalized
to probe-treated control cells to account for differences between endogenous levels of ROS/RNS
produced in cells (Henkler et al., 2010). Uncertainties associated with AUC were determined by
averaging the AUCs obtained by fitting dose-response data with each point removed
systematically, following the methodology described in Tuet et al. (2016). ROS/RNS production
was also measured for background filters and found to be within the uncertainty of control cells,
indicating that there was no evidence for significant $H_2O_2$ or HONO uptake onto seed particles
(Fig. S1). Furthermore, exposure to filter extract did not result in decreases in metabolic activity
as measured by the MTT assay for all SOA systems investigated (Fig. S2). Since results from MTT
may represent the number of viable cells present, changes in inflammatory endpoints did not likely
result from changes in the number of cells exposed (i.e. decreases in response cannot be attributed
to cell death).
For all inflammatory responses measured (levels of ROS/RNS, TNF-α, and IL-6), SOA
precursor identity and formation condition influenced the level of response, as demonstrated by
the range of values obtained from different SOA precursors and different formation conditions
(Fig. 2). Despite having a clear effect, no obvious trends were observed for each variable (precursor
or formation condition) on individual responses. This is in contrast to that observed for the
oxidative potential as measured by DTT ($OP^{WS-DTT}$) for these samples, where only precursor
identity influenced $OP^{WS-DTT}$ substantially (Tuet et al., 2017). However, this may not be surprising
as DTT is a chemical assay, which only accounts for the potential of species to participate in redox
reactions (Cho et al., 2005), whereas cellular assays account for many complicated cellular events




involved in intricate positive and negative feedback loops. Due to the considerably different
classes of compounds chosen as SOA precursors, aerosol compositional changes between different
precursors were generally larger than those between different formation conditions of the same
precursor (see Fig. 3a) (Tuet et al., 2017). DTT may only be sensitive to larger differences arising
from different precursors (i.e. a different carbon backbone), whereas cellular assays could also be
sensitive to differences between different formation conditions and chemical composition of the
same precursor. Moreover, while Tuet et al. (2017) showed that the intrinsic $OP^{WS\text{-}DTT}$ spanned a
wide range, with isoprene and naphthalene SOA generating the lowest and highest $OP^{WS\text{-}DTT}$, these
bounds were less clear for cellular responses. While isoprene and naphthalene SOA still generated
the lowest and highest inflammatory responses in general, a few exceptions exist (e.g. ROS/RNS
levels induced by pentadecane SOA formed under dry, $RO_2 + HO_2$ dominant conditions, Fig. 2).

Though no apparent trends in individual inflammatory responses were observed as a

function of SOA precursor identity or formation condition, several patterns among all three
inflammatory responses were observed for SOA precursors with similar carbon backbones.
Exposure to isoprene SOA induced the lowest levels of TNF-α and IL-6 among the aerosol systems
studied (Fig. 2). Furthermore, isoprene SOA generated from different pathways (i.e.
photooxidation under different $RO_2$ fates and reactive uptake of IEPOX) (Surratt et al., 2010; Xu
et al., 2014; Chan et al., 2010) produced similar responses for each inflammatory endpoint. These
results suggest that different isoprene SOA products (Surratt et al., 2010; Xu et al., 2014; Chan et
al., 2010) may induce similarly low inflammatory responses and are consistent with the intrinsic
$OP^{WS\text{-}DTT}$ obtained for these SOA samples, where isoprene SOA generated the lowest $OP^{WS\text{-}DTT}$
of all SOA systems studied and the $OP^{WS\text{-}DTT}$ was similar for all SOA formation conditions
explored (Tuet et al., 2017). This finding is in contrast to a previous study by Lin et al. (2016),

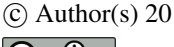



where methacrylic acid epoxide (MAE)-derived SOA was found to be substantially more potent
than IEPOX-derived SOA. However, while exposure to MAE-derived SOA induced the
upregulation of a larger number of oxidative stress response genes than IEPOX-derived SOA, the
fold change of several genes reported in Lin et al. (2016) are actually similar. Thus, it is possible
that the inflammatory cytokines measured in this study are involved in pathways concerning those
genes, resulting in a similar response level regardless of SOA formation condition.

Similarly, exposure to SOA generated from the photooxidation of $\alpha$-pinene and $m$-xylene

resulted in similar inflammatory responses for all three formation conditions (Fig. 2). These
cellular assay results are consistent with results from the DTT assay where the $OP^{WS-DTT}$ was not
significantly different between SOA formed under different formation conditions (Tuet et al.,
2017). Response levels induced by these two SOA systems are also similar across all three
inflammatory measurements investigated (Fig. 2). This suggests that products from both
precursors may induce similar cellular pathways resulting in the production of similar levels of
inflammatory markers. Indeed, there are several similarities between products formed from the
photooxidation of $\alpha$-pinene and $m$-xylene. For instance, a large portion of $\alpha$-pinene and $m$-xylene
oxidation products under both $RO_2 + HO_2$ and $RO_2 + NO$ pathways are ring-breaking products
with a similar carbon chain length (Eddingsaas et al., 2012; Vivanco and Santiago, 2010; Jenkin
et al., 2003). As a result of this similarity, products from both SOA systems may interact with the
same cellular targets and induce similar cellular pathways, resulting in a similar response
regardless of precursor identity and formation condition. These observations further imply that the
chemical structures of oxidation products may be important regardless of PM source/precursor.

A different pattern was observed for $\beta$-caryophyllene and pentadecane SOA, where the IL-

6 response spanned a much larger range than ROS/RNS and TNF-$\alpha$ (Fig. 2). This is in contrast to



the trends observed for the OP$^{WS-DTT}$ for β-caryophyllene and pentadecane SOA, where OP$^{WS-DTT}$
was similar regardless of formation condition (Tuet et al., 2017). This suggests that there are
differences between organic peroxides and organic nitrates formed from certain precursors that
influence cellular responses, but are not captured by redox potential measurements. Less is known
about the effects of humidity on SOA formation and chemical composition for all SOA systems
investigated, as most laboratory chamber studies in literature have been conducted under dry
conditions. Specifically here, very high levels of IL-6 were observed post-exposure to pentadecane
SOA formed under humid conditions. Prior studies reported opposing findings with some showing
a significant effect of water on aerosol formation and chemical composition (Nguyen et al., 2011;
Wong et al., 2015; Healy et al., 2009; Stirnweis et al., 2016), while others found little influence
(Edney et al., 2000; Boyd et al., 2015; Cocker III et al., 2001). It is clear that humidity effects are
highly hydrocarbon-dependent and further studies into the specific products formed under humid
conditions are required to understand how these differences in chemical composition may translate
to different cellular endpoints. Nonetheless, the known products formed from the photooxidation
of these hydrocarbons may provide some insight into the inflammatory responses observed. While
there are no prior studies involving pentadecane oxidation products, it is expected that the
oxidation products will be similar to those reported in the oxidation of dodecane (i.e. same
functionalities with a longer carbon chain) (Loza et al., 2014). It is therefore likely that pentadecane
oxidation products resemble long chain fatty acids and could potentially insert into the cell
membrane (Loza et al., 2014). This insertion could potentially affect membrane fluidity, which is
known to affect cell function substantially although the specific effect depends strongly on the
particular modification and cell type of interest (Baritaki et al., 2007; Spector and Yorek, 1985).
In some cases, these alterations lead to the induction of apoptosis, which involves pathways





leading to the production of TNF-α (Baritaki et al., 2007; Wang et al., 2003). TNF-α can then
induce the production of IL-6, which once produced can also inhibit the production of TNF-α in a
feedback loop (Kishimoto, 2003; Wang et al., 2003). These cellular events are consistent with the
observed inflammatory response induced by pentadecane SOA exposure, where there is a high IL-
6 response and a lower TNF-α response. The low ROS/RNS response observed is also in line with
these cellular events, as IL-6 exhibits anti-inflammatory functions, which can neutralize ROS/RNS
production. These responses are less pronounced for β-caryophyllene aerosol, which may be due
to the shorter carbon chain observed in known products (Chan et al., 2011). While β-caryophyllene
and pentadecane are both C15 precursors, β-caryophyllene is a bicyclic compound and many SOA
products retain the 4-membered ring, resulting in a shorter carbon backbone (Chan et al., 2011).
As a result, fewer products may insert into the cell membrane, leading to a lesser response
compared to pentadecane SOA exposure.

Naphthalene exhibits a completely different pattern from the rest of the SOA systems

investigated, with a large range observed for both TNF-α and IL-6 under different formation
conditions (Fig. 2). Higher levels of ROS/RNS were also observed as a result of exposure to
naphthalene aerosol irrespective of SOA formation condition. Similarly, the $OP^{WS\text{-}DTT}$ of
naphthalene SOA previously measured by Tuet et al. (2017) was an outlier among all SOA systems
investigated, as the measured $OP^{WS\text{-}DTT}$ was at least twice that of the next highest SOA system.
These observations are consistent with the formation of specific SOA products such as
naphthoquinones, which are known to induce redox-cycling in cells and are formed under both
$RO_2 + HO_2$ and $RO_2 + NO$ pathways (Henkler et al., 2010; Kautzman et al., 2010). Consequently,
aerosol generated from naphthalene may induce higher levels of inflammatory responses than
other SOA due to this process (Henkler et al., 2010; Lorentzen et al., 1979). However, as shown



by the high levels of IL-6, exposure to naphthalene SOA may also induce anti-inflammatory
pathways not captured by $OP^{WS\text{-}DTT}$ measurements. Moreover, a clear increasing trend is apparent
for TNF-α and IL-6 produced upon naphthalene SOA exposure, with a higher level of both
cytokines observed for aerosol formed under $RO_2$ + NO dominant and humid conditions.
Previously, the effect of different $RO_2$ fates on SOA $OP^{WS\text{-}DTT}$ was attributed to the different
products known to form under both pathways (Tuet et al., 2017). The same explanation applies for
cellular measurements as SOA products that promote electron transfer reactions with anti-oxidants
can result in redox imbalance as measured by $OP^{WS\text{-}DTT}$ and the induction of related cellular
pathways such as ROS/RNS and cytokine production (Tuet et al., 2017). Finally, naphthalene SOA
induced cellular responses outside of those observed for other aerosol systems, with higher levels
of all inflammatory markers than other SOA systems. As shown previously for $OP^{WS\text{-}DTT}$,
naphthalene may be an outlier due to aromatic ring-containing products, which may then induce
different cellular pathways compared to other aerosol systems investigated, the products of which
do not contain aromatic rings. Additionally, many known aerosol products formed from the
photooxidation of naphthalene have functionalities that resemble those of dinitrophenol, which is
known to decouple phosphorylation from electron transfer (Terada, 1990). It is therefore possible
that the aromatic functionality present in the majority of naphthalene SOA products results in the
involvement of very different cellular pathways, leading to outlier inflammatory endpoint
responses. Various products of naphthalene oxidation such as nitroaromatics and polyaromatics
are known to have mutagenic properties and may induce the formation of DNA adducts (Baird et
al., 2005; Helmig et al., 1992). As such, it is possible that these products may induce health effects
via other pathways as well and naphthalene SOA exposure may have effects beyond redox
imbalance and oxidative stress.





Bulk aerosol elemental ratios (O:C, H:C, and N:C) were determined for each SOA system
investigated. Different types of organic aerosol are known to span a wide range of O:C, which may
be utilized as an indication of oxidation, and the van Krevelen diagram was used to visualize
whether changes in O:C and H:C ratios corresponded to changes in levels of inflammatory
response (Fig. 3a, S3) (Chhabra et al., 2011; Lambe et al., 2011; Ng et al., 2010). Changes in the
slope within the van Krevelen space provide information on SOA functionalization (Heald et al.,
2010; Van Krevelen, 1950; Ng et al., 2011). Beginning from the precursor hydrocarbon, a slope
of 0 indicates alcohol group additions, a slope of -1 indicates carbonyl and alcohol additions on
separate carbons or carboxylic acid additions, and a slope of -2 indicates ketone or aldehyde
additions.
As seen in Fig. 3a, the laboratory-generated aerosols span a large range of O:C and H:C
ratios. Both SOA formation condition and precursor identity influenced elemental ratios, however,
precursor identity generally had a larger effect as evident by the clusters observed for different
SOA precursors. Despite these differences in chemical composition, there were no obvious trends
between O:C or H:C and any inflammatory endpoint measured. This is similar to that observed for
chemical oxidative potential as measured by DTT, where a higher O:C did not correspond to a
higher oxidative potential for both laboratory-generated and ambient aerosols (Tuet et al., 2017).
This is likely due to the different formation conditions used to generate SOA, which may not be
directly comparable. Nevertheless, a significant correlation was observed between ROS/RNS and
$\overline{OS}_c$ (Fig. 3b). This positive correlation is not surprising, as a higher average oxidation state would
likely correspond to a better oxidizing agent. Future studies should evaluate the effect of the degree
of oxidation for SOA formed from the same SOA precursor under the same formation condition
to investigate whether atmospheric aging of aerosol (which typically leads to increases in the



degree of oxidation) affects inflammatory responses. Finally, the N:C ratio was also determined
for SOA systems formed under conditions that favor the $RO_2$ + NO pathway (Fig. S4) and were
found to span a large range. Similarly, there was no obvious trend between N:C ratios and the
inflammatory endpoints measured.

**Relationship between inflammatory responses.** To visualize whether there exists a

relationship between inflammatory markers measured, levels of TNF-α and IL-6 are shown in Fig.
4, sized by ROS/RNS. With the exception of naphthalene SOA, the inflammatory cytokine
responses for all aerosol systems investigated follow an exponential curve (Fig. 4, shown in black)
where there appears to be a plateau for TNF-α levels. Along this curve, ROS/RNS levels also
appear to increase with increasing inflammatory cytokine levels to a certain point, after which
ROS/RNS levels decrease. These observations are in line with the interconnected effects of both
cytokines. While both TNF-α and IL-6 have pro-inflammatory effects that may lead to the increase
of ROS/RNS production, the individual pathways are also involved in many complicated
stimulation and inhibition loops and there is extensive cross-talk between both pathways. For
instance, TNF-α induces the production of glucocorticoids, which in turn inhibits both TNF-α and
IL-6 production (Wang et al., 2003). IL-6 also directly inhibits the production of TNF-α and other
cytokines induced as a result of TNF-α (e.g. IL-1) and stimulates pathways that lead to the
production of glucocorticoids (Kishimoto, 2003). As a result, increases in IL-6 may be
accompanied by decreases in TNF-α, resulting in the observed plateau. Furthermore, ROS/RNS
levels may represent a fine balance between anti-inflammatory and pro-inflammatory effects. Both
cytokines are involved in the acute phase reaction and can affect ROS/RNS levels via pro-
inflammatory pathways. IL-6 also exhibits some anti-inflammatory functions and may thus lower
ROS/RNS levels as well. These interconnected pathways could account for the observed parabolic





pattern for ROS/RNS production. Exposure to naphthalene SOA resulted in responses outside of
those observed for other aerosol systems, likely due to the formation of aromatic ring-retaining
products as discussed in the previous section.
**Comparison with ambient data.** To evaluate how the oxidative potential and ROS/RNS
production of the SOA systems investigated compare in the context of ambient samples, the
measurements obtained in this study were plotted with those obtained in our previous study
involving ambient samples collected around the greater Atlanta area (Fig. 5) (Tuet et al., 2016).
These ambient samples were analyzed using the same methods for determining oxidative potential
(DTT assay (Cho et al., 2005; Fang et al., 2015b)) and ROS/RNS production (cellular carboxy-
$H_2DCFDA$ assay (Tuet et al., 2016)). Furthermore, the same extraction protocol (water-soluble
extract) was followed in both studies (Tuet et al., 2016). Results from both studies are therefore
directly comparable. Previously, a significant correlation between ROS/RNS production and
oxidative potential as measured by DTT was observed for summer ambient samples. In the same
study, correlations between ROS/RNS production and organic species were also observed for
summer ambient samples, and it was suggested that these correlations may reflect contributions
from photochemically produced SOA (Tuet et al., 2016).
Fig. 5 shows that laboratory-generated SOA oxidative potential is comparable to that
observed in ambient samples, with the exception of naphthalene SOA, which produced higher
DTT activities due to its aromatic ring retaining products (Tuet et al., 2017; Kautzman et al., 2010).
Laboratory-generated SOA also induced similar or higher levels of ROS/RNS compared to
ambient samples. There are many possible explanations for the observed higher response for some
SOA samples. For instance, individual, single precursor SOA systems were considered in this
study, whereas ambient aerosol contains SOA from multiple precursors as well as other species



that are not considered in this study (e.g. metals). Interactions between SOA from different
precursors is likely to occur and may result in different response levels. Complex interactions
between SOA and other species present in the ambient (e.g. metals or other organic species) are
also likely involved (Tuet et al., 2016). Previous studies have also suggested the possibility of
metal-organic complexes. For instance, Verma et al. (2012) showed that certain metals were
retained on a C-18 column, which is utilized to remove hydrophobic components, suggesting that
these metals were likely complexed and removed in the process. Further chamber studies involving
photochemically generated SOA and metals may elucidate these interactions. Furthermore, there
are likely species present in the ambient that do not contribute to ROS/RNS production. That is,
while certain species contribute to the mass of PM, there is little to no ROS/RNS production
associated with these species. Ambient samples where these species comprise a significant fraction
will have a low per mass ROS/RNS production level. Finally, only three SOA formation conditions
were investigated in this study. There are multiple other possible oxidation mechanisms that lead
to the formation of SOA in the ambient, which were not accounted for in this study. Nonetheless,
despite the low ROS/RNS levels observed post SOA exposure, there is an association between
ROS/RNS production and DTT activity (Fig. 5). These results suggest that our previous findings
based on ambient filter samples may be extended to SOA samples. That is, while the relationship
between ROS/RNS production and DTT activity is complex, DTT may serve as a useful screening
tool as samples with low DTT activities are likely to produce low levels of RNS/RNS (Tuet et al.,

2016).

**Implications.** Levels of ROS/RNS, TNF-$\alpha$, and IL-6 were measured after exposing cells
to the water-soluble extract of SOA generated from the photooxidation of six SOA precursors
under various formation conditions. Although previous epidemiological and ambient studies have



found correlations between metals and various measures of health effects (Verma et al., 2010;
Pardo et al., 2015; Burnett et al., 2001; Huang et al., 2003; Akhtar et al., 2010; Charrier and
Anastasio, 2012), the measured levels of TNF-α, IL-6, and ROS/RNS obtained in this study
demonstrate that organic aerosols alone can induce a cellular response. This was previously
observed for the oxidative potential as measured by DTT activity as well, where the same
laboratory-generated organic aerosol samples catalyzed redox reactions and resulted in
measureable DTT decay in the absence of metal species (Tuet et al., 2017).

Results from this study also show that SOA precursor identity and formation condition

influenced response levels, with naphthalene SOA producing the highest cellular responses of the
SOA systems investigated. As discussed previously, the aromatic functionality present in many
naphthalene photooxidation products may be an important consideration for health effects. It may
therefore be worthwhile to investigate other anthropogenic aromatic ring-containing precursors as
well and to closely study the cellular effects of naphthalene SOA products given its high response.
Several patterns were also noted for SOA systems whose products shared similar functionalities
and chemical structures. For instance, photooxidation productions from pentadecane and β-
caryophyllene share similarities with long chain fatty acids and may participate in membrane
insertions, whereas many known products of naphthalene photooxidation are mutagens capable of
inducing cellular pathways beyond those that affect cellular redox balance (Baird et al., 2005;
Helmig et al., 1992). Given these observations, it may be possible to roughly predict responses
based on known SOA products as SOA systems whose products share similar functionalities and
carbon chain length (i.e. similar carbon backbone) are likely to induce similar cellular pathways
and produce similar levels of various inflammatory endpoints. Exposure studies involving
individual classes of SOA products may elucidate further details as to whether these types of



predictions would be plausible. Moreover, such studies could be used to determine whether the
hypothesized cellular pathways are indeed involved and whether certain cellular functions are
indeed affected by specific products (e.g. membrane insertion by pentadecane photooxidation
products and oxidative phosphorylation decoupling by naphthalene photooxidation products).

Mixture effects may be another important consideration as ambient PM contains SOA

formed from multiple SOA precursors. As a result, precursor emissions and their corresponding
SOA formation potential must be considered to fully assess PM health effects. Furthermore, it may
be worthwhile to investigate various prediction models for multi-component mixtures to bridge
the gap between laboratory studies and real ambient exposures. For instance, concentration
addition may not apply as ambient aerosol is formed in the presence of multiple precursors and the
SOA produced may induce response levels completely different from those observed for single
precursor SOA systems that comprise the mixture. Interactions between organic components and
metal species have also been suggested in previous studies (Verma et al., 2012; Tuet et al., 2016)
and may influence responses significantly. While these interactions were not considered in the
current study, there may be evidence to support the plausibility of mixture effects as ambient PM
samples produced lower levels of ROS/RNS than that of any single SOA system investigated.
Laboratory chambers can serve as an ideal platform to investigate mixture effects, as experiments
can be conducted under well-controlled conditions where the aerosol chemical composition and
health endpoints can be determined.

Finally, this study confirms that while there is not one simple correlation between oxidative

potential and cellular responses for different PM samples, the DTT assay may serve as a useful
screening tool as a low DTT activity will likely correspond to a low cellular response. Furthermore,
while ROS/RNS may serve as a general indicator of oxidative stress, there may be instances where



a low level of ROS/RNS does not necessary indicate a lack of cellular response. In the current
study, ROS/RNS levels were associated with levels of inflammatory cytokines for the majority of
SOA systems investigated. However, aerosol formed from the photooxidation of pentadecane
induced low levels of ROS/RNS production and relatively high levels of both cytokines (i.e. higher
than expected given the ROS/RNS level measured). These results suggest that at least one
additional measure (e.g. inflammatory cytokines) may be required to fully interpret ROS/RNS
measurements.





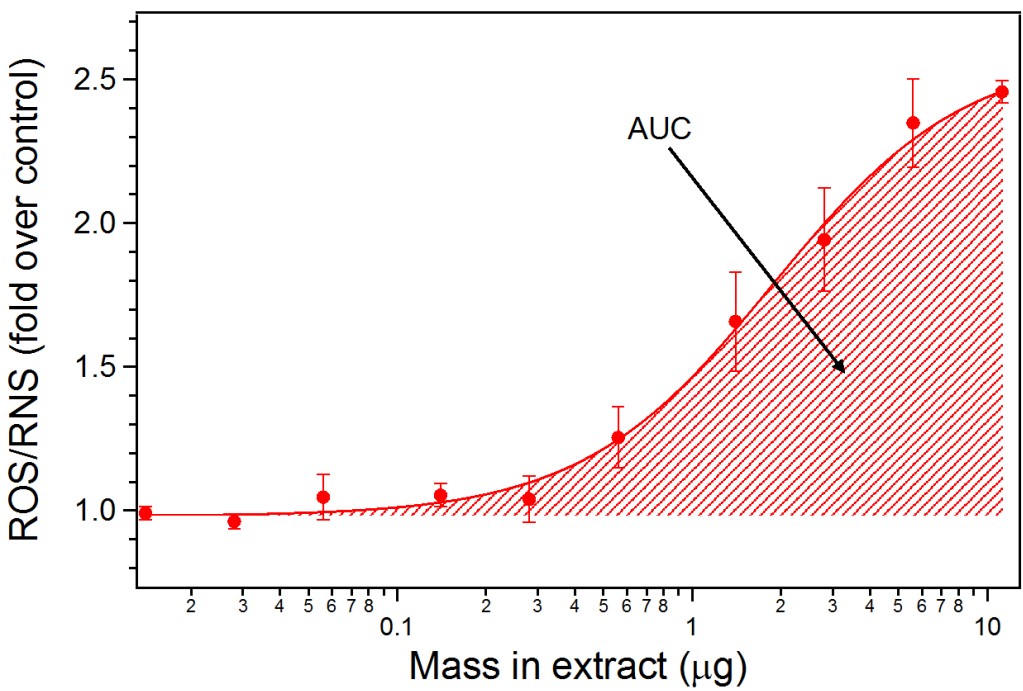


**Figure 1.** Representative dose-response curve of ROS/RNS produced as a result of filter

exposure (naphthalane SOA formed under dry, $RO_2$ + NO dominant conditions). ROS/RNS is

expressed as a fold increase over control cells, defined as probe-treated cells incubated with

stimulant-free media. Dose is expressed as mass in extract (μg). Data shown are means ±

standard error of triplicate exposure experiments. The Hill equation was used to fit the dose-

response curve and the area under the dose-response curve (AUC) is shown.

529





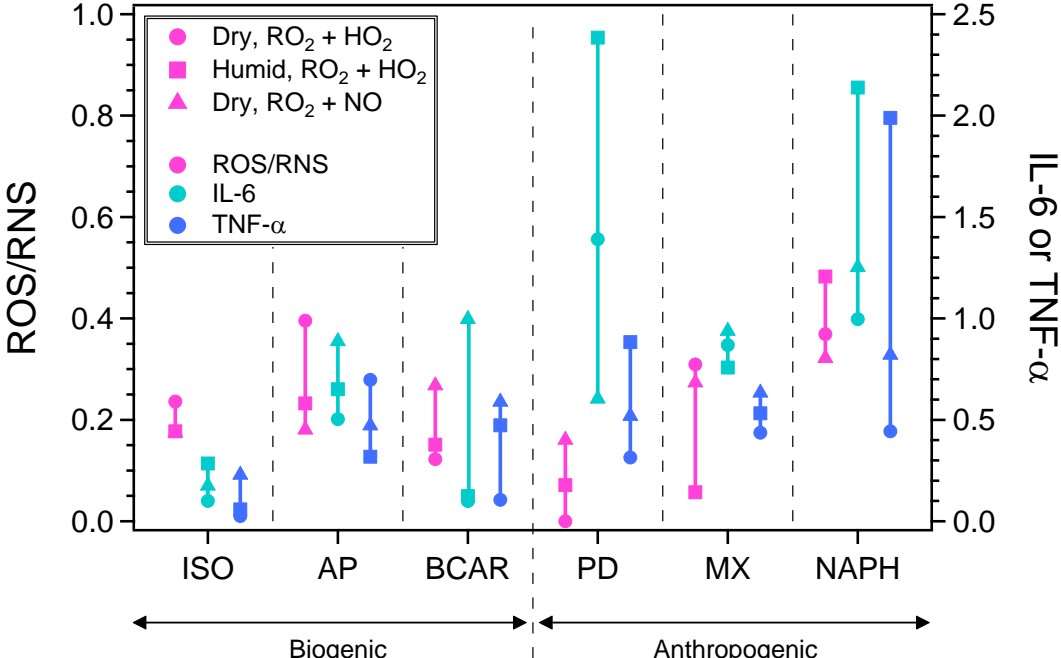

530

**Figure 2.** Area under the dose-response curve for various inflammatory responses induced as a result of SOA exposure: **ROS/RNS**, **IL-6**, and **TNF-α**. SOA were generated from various precursors (ISO: isoprene, AP: α-pinene, BCAR: β-caryophyllene, PD: pentadecane, MX: *m*-xylene, and NAPH: naphthalene) under various conditions (circles: dry, $RO_2 + HO_2$; squares: humid, $RO_2 + HO_2$; and triangles: dry, $RO_2 + NO$). Lines connecting the same inflammatory response for SOA generated from the same precursor under different formation conditions are also shown.



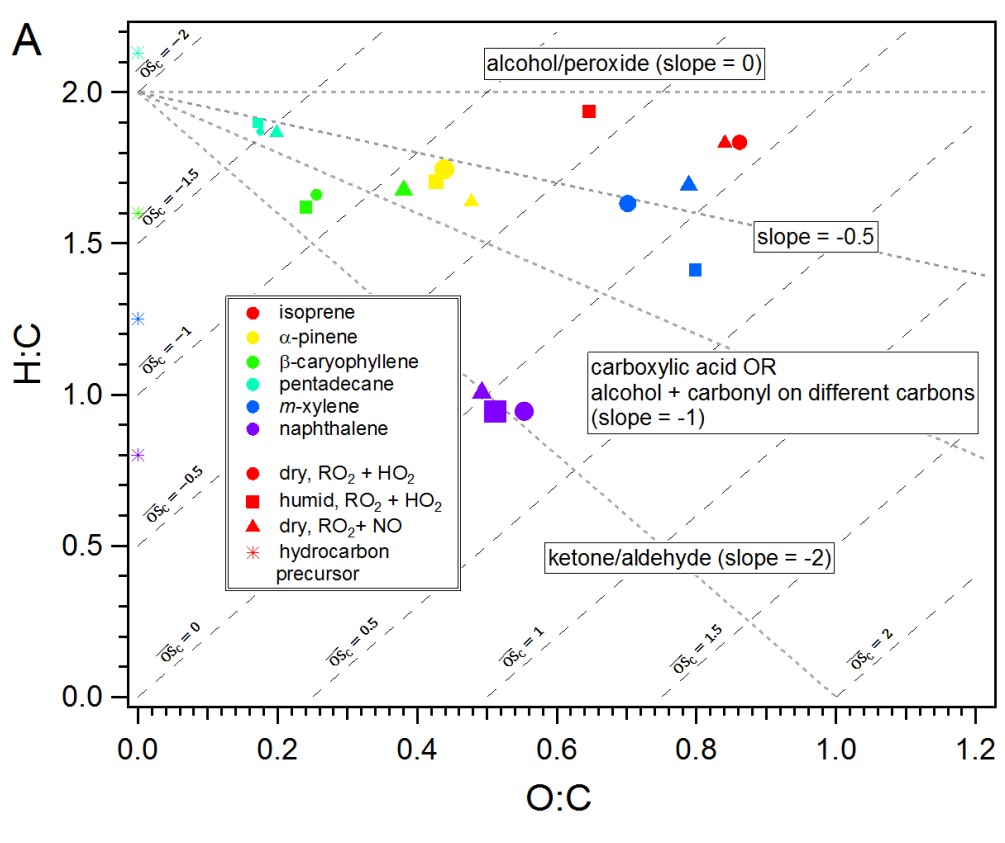

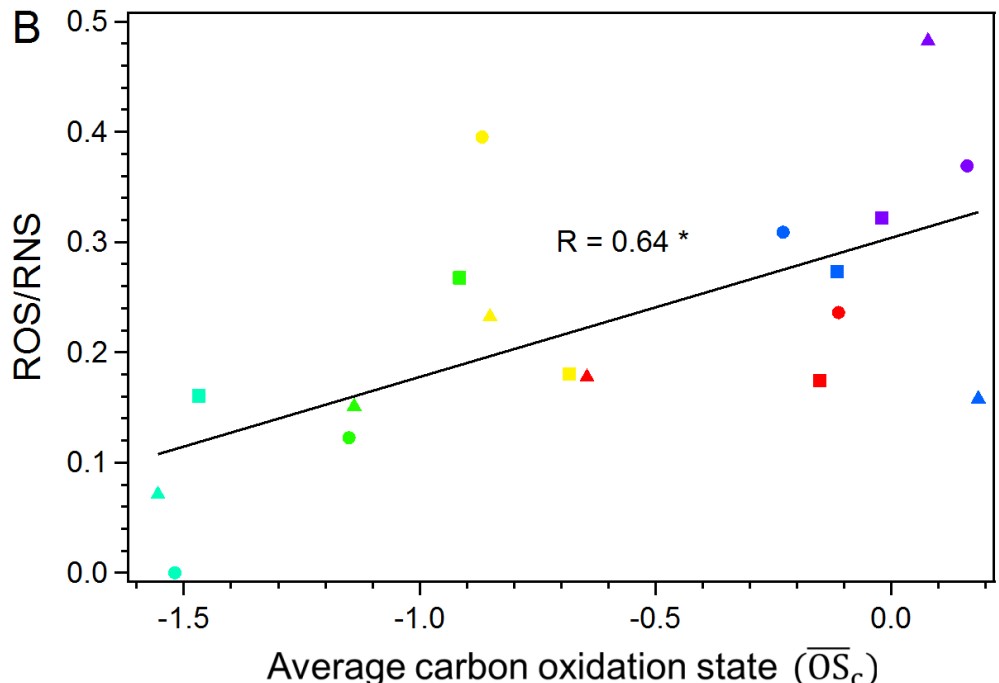





**Figure 3.** van Krevelen plot for various SOA systems sized by ROS/RNS levels (panel A) and
correlation between ROS/RNS levels and average carbon oxidation state (panel B). Data points
are colored by SOA system (red: isoprene, yellow: α-pinene, green: β-caryophyllene, light blue:
pentadecane, blue: *m*-xylene, and purple: naphthalene), shaped according to formation conditions
(circle: dry, $RO_2 + HO_2$; square: humid, $RO_2 + HO_2$; and triangle: dry, $RO_2 + NO$). SOA precursors
are shown as stars, colored by SOA system.





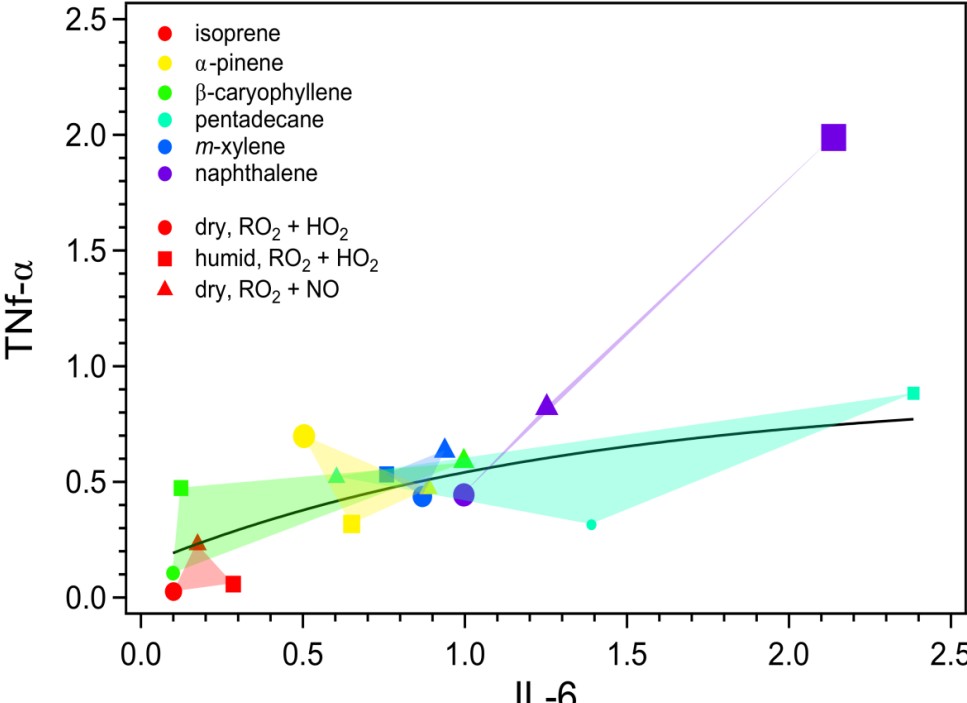


**Figure 4.** Area under the dose-response curve per mass of SOA for various inflammatory

responses induced as a result of SOA exposure. Data points are sized according to ROS/RNS level.

SOA were generated from various SOA precursors (red: isoprene, yellow: α-pinene, green: β-

caryophyllene, light blue: pentadecane, blue: *m*-xylene, and purple: naphthalene) under various

conditions (circles: dry, $RO_2 + HO_2$; squares: humid, $RO_2 + HO_2$; and triangles: dry, $RO_2 + NO$).

A fitted curve excluding naphthalene data is shown as a guide. Shaded regions for each system,

colored by SOA precursor, are also shown to show the extent of clustering and provide a

visualization for the different patterns observed.





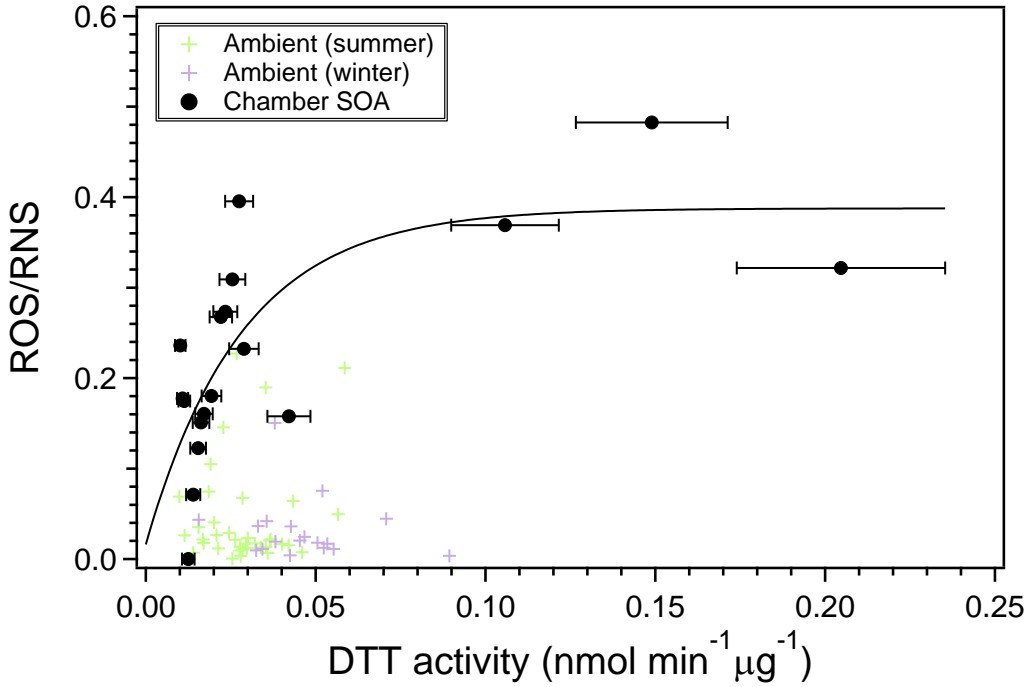

554

**Figure 5.** ROS/RNS production and intrinsic DTT activities for chamber SOA and ambient

samples collected around the greater Atlanta area. All samples were analyzed using the method

outlined in Cho et al. (2005) and Tuet et al. (2016). Ambient samples are colored by season as

determined by solstice and equinox dates between June 2012 and October 2013 (Tuet et al.,

2016). A fitted curve for laboratory-generated samples is shown as a guide.



**Table 1.** Experimental conditions.

| Experiment | SOA precursor | OH precursor | Relative humidity (%) | [HC]$_0$ (ppb) |
|---|---|---|---|---|
| 1 | isoprene | $H_2O_2$ | <5% | 97 |
| 2 | α-pinene | $H_2O_2$ | <5% | 191 |
| 3 | β-caryophyllene | $H_2O_2$ | <5% | 36 |
| 4 | pentadecane | $H_2O_2$ | <5% | 106 |
| 5 | *m*-xylene | $H_2O_2$ | <5% | 450 |
| 6 | naphthalene | $H_2O_2$ | <5% | 178 |
| 7 | isoprene | $H_2O_2$ | <5%[a] | 97 |
| 8 | α-pinene | $H_2O_2$ | 40% | 334 |
| 9 | β-caryophyllene | $H_2O_2$ | 42% | 63 |
| 10 | pentadecane | $H_2O_2$ | 45% | 106 |
| 11 | *m*-xylene | $H_2O_2$ | 45% | 450 |
| 12 | naphthalene | $H_2O_2$ | 44% | 431 |
| 13 | isoprene | HONO | <5% | 970 |
| 14 | α-pinene | HONO | <5% | 174 |
| 15 | β-caryophyllene | HONO | <5% | 21 |
| 16 | pentadecane | HONO | <5% | 74 |
| 17 | *m*-xylene | HONO | <5% | 431 |
| 18 | naphthalene | HONO | <5% | 145 |

[a] Acidic seed (8 mM $MgSO_4$ and 16 mM $H_2SO_4$) was used instead of 8 mM $(NH_4)_2SO_4$





ACKNOWLEDGMENT
This work was supported by the Health Effects Institute under research agreement No. 4943-
RFA13-2/14-4. Wing Y. Tuet acknowledges support by the National Science Foundation
Graduate Research Fellowship under Grant No. DGE-1650044.
ABBREVIATIONS
PM: particulate matter; SOA: secondary organic aerosol; ROS/RNS: reactive oxygen/nitrogen
species; TNF-α: tumor necrosis factor-α; IL-6: interleukin-6

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
