# Peer review of "Inflammatory responses to secondary organic aerosols (SOA) generated from biogenic and"

_Atmospheric Chemistry and Physics, 2017_

## Referee Comment (RC1) · Anonymous Referee #1 · 17 May 2017

In this work, the authors examined the cellular response to laboratory-generated secondary organic aerosol (SOA). The health impacts of SOA have been an emerging topic lately, and this work focuses on characterizing the production of reactive oxygen/nitrogen species (ROS/RNS) in macrophages for different types of SOA, and the production of biomarkers of oxidative stress. This is an important topic because oxidative stress has been proposed to be the main mechanism by which particulate matter leads to cardiopulmonary outcomes.

The authors produced SOA in chamber experiments under various conditions, collected SOA onto filters, and exposed SOA extracts to a cell line for murine alveolar macrophages. The results were compared to measurements of chemical oxidative

potential using the now commonly used dithiothreitol (DTT) assay. While there is a general agreement in oxidative response, there are subtle differences between the assays, suggesting different biological mechanisms that may inspire future work. The experiments are conducted in a careful manner and the results are well interpreted. This is a new area for ACPD to publish, but I believe it is a relevant topic. Otherwise the paper is technically strong and I have no major comments. It should be published after these minor comments:

- Page 4 Line 73: The authors state that there are many gaps. What are the gaps? What is the specific gap this work is attempting to address?

- Page 4 Line 76-81: the authors state that health studies focus on primary emissions rather than SOA, but then cited more SOA studies than primary studies. Seems contradictory. In fact, there is now a lot of attention on SOA. I suggest rephrasing.

- Page 5 line 95: Why were IL-6 and TNF-alpha chosen as the biomarkers? There are many other markers (such as HO-1, IL-17). Are these biomarkers better indicators of oxidative stress and better linked to health endpoints than others? Given that there is a nuanced response shown in Fig. 4, perhaps the choice of IL-6 and TNF-alpha was deliberate, but as a reader I am not sure why.

- Page 8 Line 149: 45% relative humidity is still quite dry. I would not label it as "humid".

- Page 8 lines 154-158: does an acidic seed affect the background ROS production? Or is there sufficient buffer that cells are exposed to the same pH?

- Page 8 line 161: What is zero air? Is this purified air? How is the air purified?

- Page 8 line 169: presumably this concentration of OH is yielded only upon irradiation for the specific set of chamber lights.

- Page 10 line 212: why is 24 hrs chosen? What happens if cytokine levels were measured earlier or later? Are there recovery effects of exposure?

- Page 12 line 247: H2O2 is unlikely to be taken up by inorganic seeds particles on a Teflon filter (as shown by the authors' results), but may be taken up if there are organics coated on the filter. Is it possible there is further heterogeneous reactions of H2O2 on the organics, given the H2O2 concentrations are 3ppm?

- Page 12-13 lines 259-268: This is a central finding of this manuscript: the carbon backbone seems to play a bigger role than formation conditions. While I do not dispute the results, this finding is hard to rationalize. Formation conditions will affect mostly the functional groups that go onto the molecule (there may be small changes in the backbone with fragmentation pathways), while precursor identity will determine the size and shape of the backbone. ROS is likely produced through electron transfer to/from the functional groups interacting (or reacting) with O2, H2O, antioxidants and NAPDH. It is therefore difficult to imagine that the functional group matters less than the backbone structure. Also, by that logic, reactions that change the molecular structure (such as oligomerization, fragmentation) would change the cellular ROS quite significantly. Is there any evidence of that?

- Page 16 line 326: this is an interesting explanation. If fatty acids are really changing cell functions that significantly, meat cooking organic aerosols, which are composed almost entirely of fatty acids, would elicit very strong responses.

- Page 16 Line 343: Naphthalene is not "completely" different. For example, IL-6 and TNF-alpha are still somewhat positively correlated at low levels. Perhaps it is just a more distinct pattern.

- Page 18 Line 395-396 and Fig. 3b: what does significant correlation mean? There is an asterisk in Fig. 3b. Does that mean the trend is statistically significant? If so, please provide statistical justification (e.g. 95% confidence interval?). Does it have to be a linear model? Does the correlation still stand if naphthalene SOA (which is the outlier) points are removed? It would seem reasonable to me to remove the naphthalene system if there is reason to be believe it has a very different toxicological mechanism.

- Page 19 lines 404-425: What is the relationship between ROS/RNS and cytokines for these SOA systems? It seems that plotting them against each other would help explain trends in each SOA system, or at least establish whether or not ROS/RNS are linked to upregulation of these cytokines.

Technical comments:

- Page 3 Line 52: "anti-oxidant" should be "antioxidant"

- Page 7 line 127: "form" should be "from"

- Page 21 line 464: "RNS/RNS" should be "ROS/RNS"
* * *

---

## Referee Comment (RC2) · Anonymous Referee #2 · 30 Jun 2017

Overall Recommendation:

This study examines an important aspect of atmospheric chemistry that hasn't been well examined in the past. Specifically, it remains unclear what the exact adverse effects (if any) of SOA are on human health. To begin understanding the potential adverse effects of SOA are on human health, this study exposed alveolar macrophages to SOA generated from the photooxidation of biogenic (isoprene, a-pinene, and b-caryophyllene) and anthropogenic (pentadecane, m-xylene, and napthalene precursors under varying condition of NOx level and humidity. Specific cellular responses were measured, including reactive oxygen/nitrogen species (ROS/RNS) production

and secreted levels of cytokines (TNF-a and IL-8), with the latter known to be related to inflammatory response. I think this paper will eventually be publishable in ACP, but there are a number of revisions that I outline below that need to be considered very seriously by the authors. Some of the revisions relate to justifications missing for the specific cell line used here, comparisons to other literature, and also making more stronger connections to what this all means in terms of biological mechanisms. Many of the potential biological pathways altered by the exposures don't seem to be well justified using citations to the prior toxicological literature. Some of the results (as I expand below) may need to be reanalyzed and conclusions changed based on this re-analysis. Lastly, the conclusions appear muddled and sometimes hard to understand. I think it is really important to clarify the relationship between DTT activity, oxidative stress, inflammation, and downstream health effects. Specifically, particle-bound and particle-induced ROS are not necessarily the same. At this time, I must recommend accept with major revisions noted.

Specific Comments:

1.) Limitations of this study: I didn't see any discussion regarding the limitations of this study, and they mainly cited their own DTT papers throughout the discussion. This would be my most major criticism. You have to be careful to say that your acute exposures here will really translate to the in vivo condition. Specifically, why does one need to be careful in extending the results obtained from in vitro exposures to the in vivo condition? What are the potential issues with extracting filters for resuspension into cell culture? Does the chemistry change, and if so, how might that affect the toxicological response?

2.) Rationale for using murine alveolar macrophages: I think the authors should provide the rationale for using murine alveolar macrophages for this study. Would certain phenotype of this cell line differ from human alveolar macrophages? How easily relatable is it to human cells? What are limitations of cell lines versus primary cells and would that matter?

3.) I noticed that the authors' cell culture and exposure media contain fetal bovine serum (FBS), which is known to potentially interfere with the ELISA assays. Normally people use serum-free media to avoid such interferences. Do the authors have any control experiments to show that FBS wouldn't interfere with their ELISA measurement?

4.) They use the cell media to extract filters. Since cell media contain a lot of supplementary materials/nutrients, would this affect the fraction of SOA materials extracted? Also, for the reactive products, would they be hydrolyzed before cell exposure?

5.) Lines 265-266: I think the red-ox activity is likely more sensitive to the functionality/electronic configuration of the functional groups, instead of carbon backbone. If it is carbon backbone, it looks to me that DTT is removed by other mechanisms such as absorption, but not through red-ox mechanisms.

6.) How these inflammatory responses relate to each other? Are they involved in the same biological network? They probably need to provide a more detailed biological background for the biomarkers they measured. For example, TNF-alpha induces IL-8 via NF-$\kappa$B. This is well known in the toxicological literature. In some of the toxicological literature, TNF-alpha is used as positive control to stimulate IL-8 in BEAS-2B cells. I don't see a clear connection between the endpoints they measured in this paper and this needs to be more justified. Without a connection to a specific biological system, it makes it hard (especially for an atmospheric chemist I'm sure) to understand what your results really mean.

7.) Lines 288-290: The authors cite Lin et al. (2016, ES&T Letters), but I think this discussion is really unclear. What genes are similar? What pathways do the authors mean? They should make them clear. Note that Lin et al. (2016, ES&T Letters) only measured oxidative stress-associated genes, but not inflammatory genes in that paper. I noted that Lin et al. (2017, ES&T) just had a newly accepted paper where they found most genes are associated with the Nrf2 pathway, but not much inflammatory

response from isoprene SOA exposure under non-cytotoxic conditions. Also, in Lin et al. (2017, ES&T) time course experiments, they found that IL-8 expression is time-sensitive. The expression maximized at 9 hr and much lowered at 24 hr, which was also shown in Arashiro et al. (2016, ACP). Their cellular materials were collected 24 hr post-exposure, so they might have missed the peak. How do the authors justify the 24 hr post exposure time? Did they conduct a series of time course experiments to see where things might peak in terms of cellular response. The authors and readers need to realize you may only captured 1 slice in time in how the cells responded.

8.) Line 305: what kind of chemical structure do they mean here?

9.) Line 322-329: I am not sure about the insertion of pentadecane oxidation products to the membrane. They should at least provide some references to support such a statement. I would expect some cellular response, specifically cytotoxicity, from these products since they are detergent like, which could potentially rupture the cell membrane. Did they see cell death from MTT data for pentadecane oxidation products?

10.) The mechanism of PAH-DNA adduct formation is well known through metabolic activation to diol epoxides. This is not mentioned at all in current discussion.

11.) Statistical Analysis: One more critical comment relates to the authors statistical analysis. Where are their linear regression resultss and the associated p values? Also, with multiple groups, one-way ANOVA should be used instead of student's t test to get p-values (same idea as the increasing type I error with multiple testing). Lastly, when they talked about the trend, I didn't see any statistical support to differentiate between groups. Are the results really statistically significant?

12.) I'm curious why the authors didn't gravimetrically weigh the filters before and after sampling to insure actual mass on filter for dose-response purposes? If you use the SMPS, you must make assumptions about density to calculate the mass. How was density accurately determined if you did use that approach? Was the SMPS sheath flow conditioned to the appropriate RH used in the chamber?

13.) Related to #12 above, were extraction efficiencies of aerosol mass from the filters determined by spiking them with representative internal standards? Extracting filters with cell media l may not actually remove a lot of materials (such as oligomers of SOA) from the filters. Why wasn't organic solvents used, then dried, and then the dried extracts reconstituted with cell media for the exposures? Toxicologists might find your dosing completely uncertain as its hard to gauge how well you removed the SOA from the filters without this information. This is a very important point for Figures like Figure 3. The AMS sees most of the SOA mass but filter extractions may not actually remove all of it for the exposure assessment done here.

14.) I have a curiosity question. Did the authors observe brown color on some of their filters (like from naphthalene SOA or isoprene SOA)? If so, did you seen any trends with brown carbon and your toxicological endpoints?

Minor Comments:

1.) Line 81-84: This seems to be an incomplete sentence or poorly worded sentence. Please revise.
* * *

---

## Author Comment (AC1) · 19 Jul 2017

We thank the reviewers for their time and comments. Below are detailed responses to each comment. The responses are italicized and the modified texts are in red.

Response to anonymous referee #1 comments:

1) **Page 4 Line 73: The authors state that there are many gaps. What are the gaps? What is the specific gap this work is attempting to address?**

   *Thank you for your comment. There are currently too many gaps to include a comprehensive list. The current work focuses on addressing the relative toxicities of different SOA systems, which was mentioned in lines 97 – 104. We have added an additional sentence in this section to clearly state the gap the current work is attempting to address.*

   Line 75: "Despite these findings, there are still many gaps in knowledge regarding PM-induced health effects. The current work will focus on the relative toxicities of different SOA systems, as field studies have repeatedly shown that SOA often dominate over primary aerosols (e.g., PM emitted directly from combustion engines) even in urban environments..."

2) **Page 4 Line 76-81: the authors state that health studies focus on primary emissions rather than SOA, but then cited more SOA studies than primary studies. Seems contradictory. In fact, there is now a lot of attention on SOA. I suggest rephrasing.**

   *We have rephrased this section accordingly.*

   Line 86: "Furthermore, in recent years, there have been an increasing number of studies on the health effects of SOA formed from the oxidation of emitted hydrocarbons, demonstrating their potential contribution to PM-induced health effects…"

3) **Page 5 line 95: Why were IL-6 and TNF-alpha chosen as the biomarkers? There are many other markers (such as HO-1, IL-17). Are these biomarkers better indicators of oxidative stress and better linked to health endpoints than others? Given that there is a nuanced response shown in Fig. 4, perhaps the choice of IL-6 and TNF-alpha was deliberate, but as a reader I am not sure why.**

   *We chose to measure IL-6 and TNF-α due to their central roles in cellular responses to stimuli and high production in MH-S cells. We have included a brief justification for choosing these specific biomarkers.*

   Line 106: "…cytokines indicative of the inflammatory response. TNF-α is a hallmark biomarker involved in triggering a number of cellular signaling cascades. More specifically, TNF-α is involved in the activation of NFκB, which regulates the expression of a variety of genes involved in inflammation and cell death, and the activation of protein kinases, which

regulate various signaling cascades (Witkamp and Monshouwer, 2000). IL-6 has both pro- and anti-inflammatory effects, and may directly inhibit TNF-α (Kamimura et al., 2004). Furthermore, both cytokines are produced at relatively high levels in MH-S cells, ensuring a high signal-to-noise ratio and thus reliable measurements (Matsunaga et al., 2001; Chen et al., 2007)."

4) **Page 8 Line 149: 45% relative humidity is still quite dry. I would not label it as "humid".**

*We prefer to label these experiments as "humid", as they can be considered relatively humid compared to our "dry" experiments (45% RH vs. 5% RH).*

5) **Page 8 lines 154-158: does an acidic seed affect the background ROS production? Or is there sufficient buffer that cells are exposed to the same pH?**

*It is unlikely for the acidic seed to affect background ROS/RNS production because the mass of seed per volume of media is low. Additionally, no changes in media color were observed during the extraction process. Since the cell culture media (RPMI-1640) contains phenol red, which is an indicator of pH, any significant changes in pH would result in an observable change in color. RPMI-1640 also uses a sodium bicarbonate buffer system to maintain physiological pH, so cells should be exposed to the same pH for all samples.*

6) **Page 8 line 161: What is zero air? Is this purified air? How is the air purified?**

*We have modified all instances of "zero air" to "pure air" and included how the air is purified at the first mention of pure air.*

Line 171: "Chambers were flushed with pure air (generated from AADCO, 747-14) for ~24 hrs…"

Line 183: "…passing pure air over the solution until it fully evaporated."

Line 186: "Naphthalene was injected by passing pure air over solid naphthalene flakes…"

7) **Page 8 line 169: presumably this concentration of OH is yielded only upon irradiation for the specific set of chamber lights.**

*Yes, this is the OH concentration yielded upon irradiation with the specific set of chamber lights ($jNO_2 = 0.28$ $min^{-1}$) (Boyd et al., 2015). This value is comparable to typical values for OH concentrations obtained in previous chamber studies (e.g., Eddingsaas et al., 2012; Loza et al., 2014; Ng et al., 2007; Chan et al., 2009; Chan et al., 2011).*

**8) Page 10 line 212: why is 24 hrs chosen? What happens if cytokine levels were measured earlier or later? Are there recovery effects of exposure?**

*We chose to measure both cytokines at 24 hrs to enable comparison at the same time point as ROS/RNS measurements (optimized in Tuet et al. (2016)) and because the production levels of both cytokines are relatively high at this time point for MH-S cells. Previous studies have shown that TNF-α and IL-6 production peak around 4 and 24 hrs, respectively (Haddad, 2001). Measuring at an earlier or later time point results in a decreased response, which may indicate recovery effects. We have modified the manuscript to clarify.*

Line 236: "following manufacturer's specifications (ThermoFisher). This time point was chosen to enable comparison with ROS/RNS levels (also measured at 24 hrs, optimized in Tuet et al. (2016)) and to ensure a high signal for both cytokines. Previous literature have shown that TNF-α and IL-6 production peak around 4 and 24 hrs, respectively (Haddad, 2001). However, while TNF-α production peaks earlier, the signal at 24 hrs is well above the detection limit of the assay, and previous studies have utilized this time point to measure both cytokines (Haddad, 2001; Matsunaga et al., 2001). Nonetheless, it should be noted that these measurements represent a single time point in the cellular response.…"

**9) Page 12 line 247: H2O2 is unlikely to be taken up by inorganic seeds particles on a Teflon filter (as shown by the authors' results), but may be taken up if there are organics coated on the filter. Is it possible there is further heterogeneous reactions of H2O2 on the organics, given the H2O2 concentrations are 3ppm?**

*Since $H_2O_2$ uptake by inorganic seed particles was not observed (as shown by the blank results), it is unlikely that more $H_2O_2$ was taken up by SOA given the hygroscopicity parameter values ($\kappa = 0.53$ for ammonium sulfate vs. $\kappa = 0.006 - 0.2$ for organic compounds) (Petters and Kreidenweis, 2007).*

**10) Page 12-13 lines 259-268: This is a central finding of this manuscript: the carbon backbone seems to play a bigger role than formation conditions. While I do not dispute the results, this finding is hard to rationalize. Formation conditions will affect mostly the functional groups that go onto the molecule (there may be small changes in the backbone with fragmentation pathways), while precursor identity will determine the size and shape of the backbone. ROS is likely produced through electron transfer to/from the functional groups interacting (or reacting) with O2, H2O, antioxidants and NAPDH. It is therefore difficult to imagine that the functional group matters less than the backbone structure. Also, by that logic, reactions that change the molecular structure (such as oligomerization, fragmentation) would change the cellular ROS quite significantly. Is there any evidence of that?**

*Thank you for your comment. We believe there was some confusion in this section. When we discussed the "carbon backbone", we intended for "carbon backbone" to include both the*

*carbon chain length and functionalities. Furthermore, we refer to the "carbon backbone" of oxidation products, rather than the precursor compound. We have modified the manuscript to clarify these points. We also note that the referenced section (lines 259 – 268) refers to findings from a previous study, where the chemical oxidative potentials of these SOA systems were measured (Tuet et al., 2017). In that study, precursor identity was found to influence oxidative potential more significantly than formation condition. We bring this up here to highlight potential differences between chemical and cellular assays. In the current study, both precursor identity and formation condition influenced the level of cellular response, and products with similar functionalities and carbon chain length may induce similar responses. Oligomerization and fragmentation reactions influence the O:C ratio (and hence $\overline{OS}_c$), of SOA. We did observe a correlation between $\overline{OS}_c$ and ROS/RNS production, shown in Fig. 3.*

Line 27: "…which suggests that the chemical structure (carbon chain length and functionalities) of photooxidation products may be important…"

Line 295: "DTT may only be sensitive to larger differences arising from different precursors, whereas cellular assays…"

Line 305: "…for SOA precursors whose products share similar chemical structures (i.e., similar carbon chain length and functionalities)…"

Line 529: "…SOA systems whose products share similar functionalities and carbon chain length are likely to induce…"

11) **Page 16 line 326: this is an interesting explanation. If fatty acids are really changing cell functions that significantly, meat cooking organic aerosols, which are composed almost entirely of fatty acids, would elicit very strong responses.**

*Thank you for the suggestion. We have added this as a potential implication.*

Line 380: "…lesser response compared to pentadecane SOA exposure. These observations, particularly those for pentadecane SOA, suggest that aerosols from meat cooking may have health implications, as fatty acids comprise a majority of these aerosols (Mohr et al., 2009; Rogge et al., 1991)."

12) **Page 16 Line 343: Naphthalene is not "completely" different. For example, IL-6 and TNF-alpha are still somewhat positively correlated at low levels. Perhaps it is just a more distinct pattern.**

*Thank you for the suggestion. We have modified the manuscript accordingly.*

Line 384: "Naphthalene exhibits a different, more distinct pattern compared to the rest of the SOA systems…"

13) **Page 18 Line 395-396 and Fig. 3b: what does significant correlation mean? There is an asterisk in Fig. 3b. Does that mean the trend is statistically significant? If so, please provide statistical justification (e.g. 95% confidence interval?). Does it have to be a linear model? Does the correlation still stand if naphthalene SOA (which is the outlier) points are removed? It would seem reasonable to me to remove the naphthalene system if there is reason to be believe it has a very different toxicological mechanism.**

*The method for determining statistical significance was described in the methods section. We have modified the manuscript and figure caption accordingly. The correlation does not hold if naphthalene SOA is removed. However, since other SOA systems (i.e., pentadecane and β-caryophyllene) may also participate in toxicological pathways unique to those SOA systems, we did not exclude naphthalene from the correlation. Furthermore, it is interesting that there exists a correlation between oxidation state and ROS/RNS even though different toxicological mechanisms may be involved.*

Line 436: "Nevertheless, a significant correlation $(p < 0.05)$ was observed…"

Line 592: "…colored by SOA system. * indicates significance, $p < 0.05$."

14) **Page 19 lines 404-425: What is the relationship between ROS/RNS and cytokines for these SOA systems? It seems that plotting them against each other would help explain trends in each SOA system, or at least establish whether or not ROS/RNS are linked to upregulation of these cytokines.**

*We show the relationship between ROS/RNS and cytokines in Fig. 4, where the ROS/RNS level is influenced by a balance between both cytokines due to pro- and anti-inflammatory effects. We did plot ROS/RNS against cytokine measurements, however, individual correlation plots did not reveal any additional information as the inflammatory markers are involved in pathways with many overlaps and crosstalk. These relationships were only apparent when all three measurements were plotted, as shown in Fig. 4.*

15) **Page 3 Line 52: "anti-oxidant" should be "antioxidant"**

*We have modified the manuscript accordingly.*

Line 52: "…redox reactions using an antioxidant species…"

Line 53: "The antioxidant is oxidized…"

Line 401: "…products that promote electron transfer reactions with antioxidants…"

**16) Page 7 line 127: "form" should be "from"**

*We have modified the manuscript accordingly.*

Line 149: "SOA formed from the photooxidation…"

**17) Page 21 line 464: "RNS/RNS" should be "ROS/RNS"**

*We have modified the manuscript accordingly.*

Line 505: "…produce low levels of ROS/RNS…"

**References:**

[revised manuscript text omitted]

---

## Author Comment (AC2) · 19 Jul 2017

We thank the reviewers for their time and comments. Below are detailed responses to each comment. The responses are italicized and the modified texts are in red. The main comments have been addressed by including a discussion on the limitations of this study and by clarifying our statistical analysis method. The revisions do not affect the conclusions of the manuscript.

Response to anonymous referee #2 comments:

1) **Limitations of this study: I didn't see any discussion regarding the limitations of this study, and they mainly cited their own DTT papers throughout the discussion. This would be my most major criticism. You have to be careful to say that your acute exposures here will really translate to the in vivo condition. Specifically, why does one need to be careful in extending the results obtained from in vitro exposures to the in vivo condition? What are the potential issues with extracting filters for resuspension into cell culture? Does the chemistry change, and if so, how might that affect the toxicological response?**

   *Thank you for your suggestion. We are aware that there are limitations regarding all health studies and have modified the manuscript to include several examples of these limitations. We note that the main objective of this study was to provide perspective on the relative toxicities of different SOA systems. Further studies are required to establish whether results from in vitro assays represent in vivo animal exposures, and from there, whether results from animal exposure studies can be generalized to actual human exposures.*

   Line 552: "Additionally, this study confirms…"

   Line 562: "…to fully interpret ROS/RNS measurements. Finally, several limitations must be considered before generalizing results from this study to *in vivo* exposures. For instance, only one cell type was explored in this study, whereas an organism consists of multiple tissues comprised of multiple cell types. Interactions between different cell types and tissue systems were not considered in this study. Furthermore, the doses investigated may not fully represent real world exposures due to differences in exposure routes and potential recovery from doses due to clearance. Nevertheless, this study provides perspective on the relative toxicities of different SOA systems which future studies can build upon."

2) **Rationale for using murine alveolar macrophages: I think the authors should provide the rationale for using murine alveolar macrophages for this study. Would certain phenotype of this cell line differ from human alveolar macrophages? How easily relatable is it to human cells? What are limitations of cell lines versus primary cells and would that matter?**

   *Thank you for your comment. We have included rationale for using this cell type in the manuscript. We chose murine alveolar macrophages as they are the first line of defense*

*against environmental insults, and the particular cell line (MH-S) retains many properties of primary alveolar macrophages (e.g., phagocytosis, cytokine production, ROS/RNS production) (Sankaran and Herscowitz, 1995; Mbawuike and Herscowitz, 1989). Furthermore, we have successfully utilized this cell line to investigate the production of ROS/RNS as a result of exposure to ambient PM samples (Tuet et al., 2016). To our knowledge, immortalized human alveolar macrophages do not exist. Mice have also been widely used as a model organism for studying human responses (Rosenthal and Brown, 2007; Takao and Miyakawa, 2015). As for the choice between cell lines and primary cells, primary cells are harvested from multiple animals, which increases the response variability. Results may therefore be less reproducible compared to cell lines.*

Line 137: "Exposures were conducted using immortalized murine alveolar macrophages (MH-S, ATCC®CRL-2019™) as they are the first line of defense against environmental insults (Oberdörster, 1993; Oberdörster et al., 1992). The particular cell line also retains many properties of primary alveolar macrophages, including phagocytosis as well as the production of ROS/RNS and cytokines (Sankaran and Herscowitz, 1995; Mbawuike and Herscowitz, 1989). MH-S cells were cultured…"

3) **I noticed that the authors' cell culture and exposure media contain fetal bovine serum (FBS), which is known to potentially interfere with the ELISA assays. Normally people use serum-free media to avoid such interferences. Do the authors have any control experiments to show that FBS wouldn't interfere with their ELISA measurement?**

*We normalized all ELISA responses to a control (cell culture supernatant from cells exposed to stimulant-free media supplemented with FBS) to capture any interferences. For our time point (24 hrs), FBS supplemented media is necessary to prevent serum starvation, which is known to induce oxidative stress (Kuznetsov et al., 2011; Wright et al., 2012). We also disagree that serum-free media is generally used for ELISA measurements, as many previous studies have performed exposures using supplemented media (e.g., Mukherjee et al., 2009; Chen et al., 2007; Sullivan et al., 2000).*

4) **They use the cell media to extract filters. Since cell media contain a lot of supplementary materials/nutrients, would this affect the fraction of SOA materials extracted? Also, for the reactive products, would they be hydrolyzed before cell exposure?**

*For oxidative potential measurements, it is known that using different extraction methods (e.g., different solvent, filtration, removing the filter) results in different components extracted and hence yields different oxidative potential measurements (Gao et al., 2017). However, there are limitations for each method. For instance, using an organic solvent requires the subsequent removal of the solvent via evaporation, which may result in loss of unstable components (e.g., semi-volatile organics). In this study, we chose to adapt an extraction method best suited for cellular exposure. While media contains species that*

*would indeed alter the fraction of material extracted, these species are also present in the alveolar fluid and the extract obtained is biologically relevant. We would also like to note that plain media (without FBS) was used for extraction and that FBS was supplemented after filtration of extracts. We did not investigate the hydrolysis of reactive products due to extraction, however this would be a potential issue for all extraction methods used in offline analysis. Further studies comparing offline and online analysis are required to investigate this.*

5) **Lines 265-266: I think the redox activity is likely more sensitive to the functionality/electronic configuration of the functional groups, instead of carbon backbone. If it is carbon backbone, it looks to me that DTT is removed by other mechanisms such as absorption, but not through redox mechanisms.**

*The referenced section refers to a previous study, where the chemical oxidative potentials as determined by DTT consumption were measured for these SOA systems (Tuet et al., 2017). In this study, we focus on the cellular responses and we find that the precursor identity and formation condition are both important and affect the cellular responses significantly. We note that there may have been some confusion in this section, as we intended "carbon backbone" to include both carbon chain length and functionalities. We have modified the manuscript to clarify our findings.*

Line 27: "…which suggests that the chemical structure (carbon chain length and functionalities) of photooxidation products may be important…"

Line 295: "DTT may only be sensitive to larger differences arising from different precursors, whereas cellular assays…"

Line 305: "…for SOA precursors whose products share similar chemical structures (i.e., similar carbon chain length and functionalities)…"

Line 529: "…SOA systems whose products share similar functionalities and carbon chain length are likely to induce…"

6) **How these inflammatory responses relate to each other? Are they involved in the same biological network? They probably need to provide a more detailed biological background for the biomarkers they measured. For example, TNF-alpha induces IL-8 via NF-κB. This is well known in the toxicological literature. In some of the toxicological literature, TNF-alpha is used as positive control to stimulate IL-8 in BEAS-2B cells. I don't see a clear connection between the endpoints they measured in this paper and this needs to be more justified. Without a connection to a specific biological system, it makes it hard (especially for an atmospheric chemist I'm sure) to understand what your results really mean.**

*Thank you for your suggestion. We have included justification on our cytokine measurements.*

Line 106: "…cytokines indicative of the inflammatory response. TNF-α is a hallmark biomarker involved in triggering a number of cellular signaling cascades. More specifically, TNF-α is involved in the activation of NFκB, which regulates the expression of a variety of genes involved in inflammation and cell death, and the activation of protein kinases, which regulate various signaling cascades (Witkamp and Monshouwer, 2000). IL-6 has both pro- and anti-inflammatory effects, and may directly inhibit TNF-α (Kamimura et al., 2004). Furthermore, both cytokines are produced at relatively high levels in MH-S cells, ensuring a high signal-to-noise ratio and thus reliable measurements (Matsunaga et al., 2001; Chen et al., 2007)."

7) **Lines 288-290: The authors cite Lin et al. (2016, ES&T Letters), but I think this discussion is really unclear. What genes are similar? What pathways do the authors mean? They should make them clear. Note that Lin et al. (2016, ES&T Letters) only measured oxidative stress-associated genes, but not inflammatory genes in that paper. I noted that Lin et al. (2017, ES&T) just had a newly accepted paper where they found most genes are associated with the Nrf2 pathway, but not much inflammatory response from isoprene SOA exposure under non-cytotoxic conditions. Also, in Lin et al. (2017, ES&T) time course experiments, they found that IL-8 expression is time sensitive. The expression maximized at 9 hr and much lowered at 24 hr, which was also shown in Arashiro et al. (2016, ACP). Their cellular materials were collected 24 hr post-exposure, so they might have missed the peak. How do the authors justify the 24 hr post exposure time? Did they conduct a series of time course experiments to see where things might peak in terms of cellular response? The authors and readers need to realize you may only captured 1 slice in time in how the cells responded.**

*We have modified the manuscript to clarify this discussion. Specifically, we include an example of a gene whose fold change was similar between the two types of SOA studies in Lin et al. (2016) and discuss how that gene is related to the inflammatory cytokines measured in this study. Oxidative stress plays a crucial role in the inflammatory process, and as such, the oxidative stress related genes measured in Lin et al. (2016) may influence cytokine production. We thank the reviewer for pointing out Lin et al. (2017) and have cited the paper accordingly. We are aware that cytokine production peaks at different time points for different cytokines. In our case, TNF-α peaks around 4 hrs, while IL-6 peaks much later at 24 hrs (Haddad, 2001). We chose to measure both cytokines at the latter time point to allow comparison. Previous studies have shown that the level of TNF-α is sufficiently high at the latter time point for accurate determination (Haddad, 2001; Matsunaga et al., 2001). The manuscript has been modified to include this justification as well.*

Line 93: "However, the cellular exposure studies involving SOA focused on SOA formed from a single precursor and included different measures of response (e.g. ROS/RNS,

inflammatory biomarkers, gene expression, etc.) (Arashiro et al., 2016; Lund et al., 2013; McDonald et al., 2010; McDonald et al., 2012; Baltensperger et al., 2008; Lin et al., 2017)."

Line 318: "…the fold change of several genes reported in Lin et al. (2016) are actually similar (e.g., *ALOX12, NQO1*). Several of these genes directly affect the production of inflammatory cytokines measured in this study. For instance, studies have observed that arachidonate 12-lipoxygenase (*ALOX12*) products induce the production of both TNF-α and IL-6 in macrophages (Wen et al., 2007). As such, a similar response level regardless of SOA formation condition may be observed depending on the biological endpoints measured. Thus, it is possible that the inflammatory cytokines measured in this study are involved in pathways concerning those genes, resulting in a similar response level regardless of SOA formation condition."

Line 231: "following manufacturer's specifications (ThermoFisher). This time point was chosen to enable comparison with ROS/RNS levels (also measured at 24 hrs, optimized in Tuet et al. (2016)) and to ensure a high signal for both cytokines. Previous literature have shown that TNF-α and IL-6 production peak around 4 and 24 hrs, respectively (Haddad, 2001). However, while TNF-α production peaks earlier, the signal at 24 hrs is well above the detection limit of the assay, and previous studies have utilized this time point to measure both cytokines (Haddad, 2001; Matsunaga et al., 2001). Nonetheless, it should be noted that these measurements represent a single time point in the cellular response.…"

8) **Line 305: what kind of chemical structure do they mean here?**

*Thank you for the comment. We have modified the manuscript to clarify.*

Line 340: "These observations further imply that the chemical structures (e.g., carbon chain lengths and functionalities) of oxidation products…"

9) **Line 322-329: I am not sure about the insertion of pentadecane oxidation products to the membrane. They should at least provide some references to support such a statement. I would expect some cellular response, specifically cytotoxicity, from these products since they are detergent like, which could potentially rupture the cell membrane. Did they see cell death from MTT data for pentadecane oxidation products?**

*Thank you for the suggestion. We have included references to support this hypothesis. We did not observe decreases in cellular metabolic activity as measured by the MTT assay (mentioned in lines 282 – 286 in the revised manuscript).*

Line 363: "…could potentially insert into the cell membrane (Loza et al., 2014), as previous studies have shown that fatty acids can feasibly insert into the cell membrane bilayer (Khmelinskaia et al., 2014; Cerezo et al., 2011)."

10) **The mechanism of PAH-DNA adduct formation is well known through metabolic activation to diol epoxides. This is not mentioned at all in current discussion.**

*We mentioned the formation of DNA adducts briefly in the section on naphthalene SOA (lines 420 – 424). The specific mechanism by which these adducts are formed is beyond the scope of this study, but would be interesting to investigate in future studies.*

11) **Statistical Analysis: One more critical comment relates to the authors statistical analysis. Where are their linear regression results and the associated p values? Also, with multiple groups, one-way ANOVA should be used instead of student's t test to get p-values (same idea as the increasing type I error with multiple testing). Lastly, when they talked about the trend, I didn't see any statistical support to differentiate between groups. Are the results really statistically significant?**

*Based on the reviewer's comment, we believe the trend referenced refers to Fig. 3. The Pearson's correlation coefficient is given in the original figure. For clarity, we have modified the manuscript and figure caption to reflect that correlations were evaluated using a 95% confidence interval. Since only two variables (cellular response and bulk aerosol composition, e.g., ROS/RNS and $\overline{OS}_c$) were tested, the student's t-test and one-way ANOVA are actually equivalent (Park, 2009).*

Line 436: "Nevertheless, a significant correlation $(p < 0.05)$ was observed…"

Line 592: "…colored by SOA system. * indicates significance, $p < 0.05$."

12) **I'm curious why the authors didn't gravimetrically weigh the filters before and after sampling to insure actual mass on filter for dose-response purposes? If you use the SMPS, you must make assumptions about density to calculate the mass. How was density accurately determined if you did use that approach? Was the SMPS sheath flow conditioned to the appropriate RH used in the chamber?**

*Mass loadings were low for isoprene and pentadecane SOA. To be consistent, we choose to determine mass by integrating the SMPS volume concentrations for all SOA systems. An aerosol density of 1 g cm$^{-3}$ was assumed to facilitate comparison between studies, since SOA density varies with precursor identity and formation condition. We have added this clarification to the manuscript. For all experiments, the SMPS was connected to the chamber for 2-3 hrs before the start of the experiment to condition the recirculating sheath flow.*

Line 201: "…multiplying by the total volume of air collected. SMPS volume concentrations were converted to mass concentrations by assuming a density of 1 g cm$^{-3}$ to facilitate comparison between studies…"

13) **Related to #12 above, were extraction efficiencies of aerosol mass from the filters determined by spiking them with representative internal standards? Extracting filters with cell media l may not actually remove a lot of materials (such as oligomers of SOA) from the filters. Why wasn't organic solvents used, then dried, and then the dried extracts reconstituted with cell media for the exposures? Toxicologists might find your dosing completely uncertain as its hard to gauge how well you removed the SOA from the filters without this information. This is a very important point for Figures like Figure 3. The AMS sees most of the SOA mass but filter extractions may not actually remove all of it for the exposure assessment done here.**

*Extraction efficiencies were not measured in this study. While different extraction methods are known to result in different constituents being extracted from the PM sample, there are limitations for each method. These are discussed in a recent publication by Gao et al. (2017). For example, using an organic solvent and drying the extract for reconstitution may result in loss of unstable constituents. For this study, we chose an extraction method best suited for cellular exposure.*

14) **I have a curiosity question. Did the authors observe brown color on some of their filters (like from naphthalene SOA or isoprene SOA)? If so, did you seen any trends with brown carbon and your toxicological endpoints?**

*We only observed brown color on our naphthalene SOA filters. We did not measure brown carbon in this study.*

15) **Line 81-84: This seems to be an incomplete sentence or poorly worded sentence. Please revise.**

*Thank you for your comment. We have modified the sentence.*

Line 91: "However, the cellular exposure studies involving SOA focused on SOA formed from a single precursor and included different measures of response…"

**References:**

Arashiro, M., Lin, Y. H., Sexton, K. G., Zhang, Z., Jaspers, I., Fry, R. C., Vizuete, W. G., Gold, A., and Surratt, J. D.: In Vitro Exposure to Isoprene-Derived Secondary Organic Aerosol by Direct Deposition and its Effects on COX-2 and IL-8 Gene Expression, Atmos. Chem. Phys. Discuss., 2016, 1-29, 10.5194/acp-2016-371, 2016.

Baltensperger, U., Dommen, J., Alfarra, R., Duplissy, J., Gaeggeler, K., Metzger, A., Facchini, M. C., Decesari, S., Finessi, E., Reinnig, C., Schott, M., Warnke, J., Hoffmann, T., Klatzer, B., Puxbaum, H., Geiser, M., Savi, M., Lang, D., Kalberer, M., and Geiser, T.: Combined determination of the chemical composition and of health effects of secondary organic aerosols: The POLYSOA project, J. Aerosol Med. Pulm. Drug Deliv., 21, 145-154, 10.1089/jamp.2007.0655, 2008.

Cerezo, J., Zúñiga, J., Bastida, A., Requena, A., and Cerón-Carrasco, J. P.: Atomistic Molecular Dynamics Simulations of the Interactions of Oleic and 2-Hydroxyoleic Acids with Phosphatidylcholine Bilayers, The Journal of Physical Chemistry B, 115, 11727-11738, 10.1021/jp203498x, 2011.

Chen, C. Y., Peng, W. H., Tsai, K. D., and Hsu, S. L.: Luteolin suppresses inflammation-associated gene expression by blocking NF-kappa B and AP-1 activation pathway in mouse alveolar macrophages, Life Sci., 81, 1602-1614, 10.1016/j.lfs.2007.09.028, 2007.

Gao, D., Fang, T., Verma, V., Zeng, L., and Weber, R.: A method for measuring total aerosol oxidative potential (OP) with the dithiothreitol (DTT) assay and comparisons between an urban and roadside site of water-soluble and total OP, Atmos. Meas. Tech. Discuss., 2017, 1-25, 10.5194/amt-2017-70, 2017.

Haddad, J. J.: L-buthionine-(S,R)-sulfoximine, an irreversible inhibitor of gamma-glutamylcysteine synthetase, augments LPS-mediated pro-inflammatory cytokine biosynthesis: evidence for the implication of an I kappa B-alpha/NF-kappa B insensitive pathway, Eur. Cytokine Netw., 12, 614-624, 2001.

Kamimura, D., Ishihara, K., and Hirano, T.: IL-6 signal transduction and its physiological roles: the signal orchestration model, in: Reviews of Physiology, Biochemistry and Pharmacology, Springer Berlin Heidelberg, Berlin, Heidelberg, 1-38, 2004.

Khmelinskaia, A., Ibarguren, M., de Almeida, R. F. M., López, D. J., Paixão, V. A., Ahyayauch, H., Goñi, F. M., and Escribá, P. V.: Changes in Membrane Organization upon Spontaneous Insertion of 2-Hydroxylated Unsaturated Fatty Acids in the Lipid Bilayer, Langmuir, 30, 2117-2128, 10.1021/la403977f, 2014.

Kuznetsov, A. V., Kehrer, I., Kozlov, A. V., Haller, M., Redl, H., Hermann, M., Grimm, M., and Troppmair, J.: Mitochondrial ROS production under cellular stress: comparison of different detection methods, Analytical and Bioanalytical Chemistry, 400, 2383-2390, 10.1007/s00216-011-4764-2, 2011.

Lin, Y.-H., Arashiro, M., Martin, E., Chen, Y., Zhang, Z., Sexton, K. G., Gold, A., Jaspers, I., Fry, R. C., and Surratt, J. D.: Isoprene-Derived Secondary Organic Aerosol Induces the Expression of Oxidative Stress Response Genes in Human Lung Cells, Environmental Science & Technology Letters, 3, 250-254, 10.1021/acs.estlett.6b00151, 2016.

Lin, Y.-H., Arashiro, M., Clapp, P. W., Cui, T., Sexton, K. G., Vizuete, W., Gold, A., Jaspers, I., Fry, R. C., and Surratt, J. D.: Gene Expression Profiling in Human Lung Cells Exposed to Isoprene-Derived Secondary Organic Aerosol, Environmental Science & Technology, 10.1021/acs.est.7b01967, 2017.

Lund, A. K., Doyle-Eisele, M., Lin, Y. H., Arashiro, M., Surratt, J. D., Holmes, T., Schilling, K. A., Seinfeld, J. H., Rohr, A. C., Knipping, E. M., and McDonald, J. D.: The effects of alpha-pinene versus toluene-derived secondary organic aerosol exposure on the expression of markers associated with vascular disease, Inhal. Toxicol., 25, 309-324, 10.3109/08958378.2013.782080, 2013.

Matsunaga, K., Klein, T. W., Friedman, H., and Yamamoto, Y.: Involvement of Nicotinic Acetylcholine Receptors in Suppression of Antimicrobial Activity and Cytokine Responses of Alveolar Macrophages to Legionella pneumophila Infection by Nicotine, The Journal of Immunology, 167, 6518-6524, 10.4049/jimmunol.167.11.6518, 2001.

Mbawuike, I. N., and Herscowitz, H. B.: MH-S, a murine alveolar macrophage cell line: morphological, cytochemical, and functional characteristics, Journal of Leukocyte Biology, 46, 119-127, 1989.

McDonald, J. D., Doyle-Eisele, M., Campen, M. J., Seagrave, J., Holmes, T., Lund, A., Surratt, J. D., Seinfeld, J. H., Rohr, A. C., and Knipping, E. M.: Cardiopulmonary response to inhalation of biogenic secondary organic aerosol, Inhal. Toxicol., 22, 253-265, 10.3109/08958370903148114, 2010.

McDonald, J. D., Doyle-Eisele, M., Kracko, D., Lund, A., Surratt, J. D., Hersey, S. P., Seinfeld, J. H., Rohr, A. C., and Knipping, E. M.: Cardiopulmonary response to inhalation of secondary organic aerosol

derived from gas-phase oxidation of toluene, Inhal. Toxicol., 24, 689-697, 10.3109/08958378.2012.712164, 2012.

Mukherjee, S., Chen, L.-Y., Papadimos, T. J., Huang, S., Zuraw, B. L., and Pan, Z. K.: Lipopolysaccharide-driven Th2 Cytokine Production in Macrophages Is Regulated by Both MyD88 and TRAM, J. Biol. Chem., 284, 29391-29398, 10.1074/jbc.M109.005272, 2009.

Oberdörster, G., Ferin, J., Gelein, R., Soderholm, S. C., and Finkelstein, J.: Role of the alveolar macrophage in lung injury: studies with ultrafine particles, Environmental Health Perspectives, 97, 193-199, 1992.

Oberdörster, G.: Lung Dosimetry: Pulmonary Clearance of Inhaled Particles, Aerosol Sci. Technol., 18, 279-289, 10.1080/02786829308959605, 1993.

Park, H. M.: Comparing group means: t-tests and one-way ANOVA using Stata, SAS, R, and SPSS, 2009.

Rosenthal, N., and Brown, S.: The mouse ascending: perspectives for human-disease models, Nat Cell Biol, 9, 993-999, 2007.

Sankaran, K., and Herscowitz, H. B.: Phenotypic and functional heterogeneity of the murine alveolar macrophage-derived cell line MH-S, Journal of Leukocyte Biology, 57, 562-568, 1995.

Sullivan, K. E., Cutilli, J., Piliero, L. M., Ghavimi-Alagha, D., Starr, S. E., Campbell, D. E., and Douglas, S. D.: Measurement of Cytokine Secretion, Intracellular Protein Expression, and mRNA in Resting and Stimulated Peripheral Blood Mononuclear Cells, Clinical and Diagnostic Laboratory Immunology, 7, 920-924, 2000.

Takao, K., and Miyakawa, T.: Genomic responses in mouse models greatly mimic human inflammatory diseases, Proceedings of the National Academy of Sciences, 112, 1167-1172, 10.1073/pnas.1401965111, 2015.

Tuet, W. Y., Fok, S., Verma, V., Tagle Rodriguez, M. S., Grosberg, A., Champion, J. A., and Ng, N. L.: Dose-dependent intracellular reactive oxygen and nitrogen species (ROS/RNS) production from particulate matter exposure: comparison to oxidative potential and chemical composition, Atmos. Environ., 144, 335-344, http://dx.doi.org/10.1016/j.atmosenv.2016.09.005, 2016.

Tuet, W. Y., Chen, Y., Xu, L., Fok, S., Gao, D., Weber, R. J., and Ng, N. L.: Chemical oxidative potential of secondary organic aerosol (SOA) generated from the photooxidation of biogenic and anthropogenic volatile organic compounds, Atmos. Chem. Phys., 17, 839-853, 10.5194/acp-17-839-2017, 2017.

Wen, Y., Gu, J., Chakrabarti, S. K., Aylor, K., Marshall, J., Takahashi, Y., Yoshimoto, T., and Nadler, J. L.: The Role of 12/15-Lipoxygenase in the Expression of Interleukin-6 and Tumor Necrosis Factor-α in Macrophages, Endocrinology, 148, 1313-1322, 10.1210/en.2006-0665, 2007.

Witkamp, R., and Monshouwer, M.: Signal transduction in inflammatory processes, current and future therapeutic targets: A mini review, Veterinary Quarterly, 22, 11-16, 10.1080/01652176.2000.9695016, 2000.

Wright, C. J., Agboke, F., Muthu, M., Michaelis, K. A., Mundy, M. A., La, P., Yang, G., and Dennery, P. A.: Nuclear Factor-κB (NF-κB) Inhibitory Protein IκBβ Determines Apoptotic Cell Death following Exposure to Oxidative Stress, J. Biol. Chem., 287, 6230-6239, 10.1074/jbc.M111.318246, 2012.